# Sex-specific lipidomic signatures in aortic valve disease reflect differential fibro-calcific progression

Patricia Prabutzki [1,5], Michele Wölk [2,5], Julia Böttner[3], Zhixu Ni[2], Sarah Werner[3], Holger Thiele [3], Jürgen Schiller [1], Petra Büttner[3], Florian Schlotter[3,4] ✉ & Maria Fedorova [2] ✉

Fibro-calcific aortic valve disease (FCAVD) is the most common valvular heart disease manifesting in pathological remodeling of the aortic valve (AV) leaflets, ultimately leading to aortic stenosis. Although dyslipidemia is a driver of FCAVD pathogenesis, the precise lipidome-wide changes underlying AV fibrosis and calcification remain largely unknown. Here, we performed deep quantitative lipidomics to profile the metabolic trajectories in human tricuspid and bicuspid AVs, and found stage-dependent extrinsic and intrinsic lipid trends. Furthermore, lipids derived from infiltrating lipoproteins are further metabolized within the AV. Intrinsic lipid remodeling suggested tissue degeneration with a loss of phosphatidylserines. Surprisingly, male and female patients showed markedly different lipid signatures of FCAVD progression, with female patients accumulating significantly higher levels of sphingomyelins and ceramides. The high extent of sexual dimorphism in the valve lipidome strongly suggests that tailored approaches should be undertaken to enhance mechanistic insight and to facilitate pharmacological intervention for FCAVD.

Aortic stenosis (AS) is a severe disease, which is characterized by thickening of the aortic valve (AV) leaflets and fibro-calcific remodeling of the tissue, ultimately leading to heart failure and high mortality[1,2]. Fibro-calcific aortic valve disease (FCAVD) is the most common valvular heart disease in the aging population[3]. Surgical aortic valve replacement or transcatheter aortic valve implantation remain the only treatment options and pose a major economic burden on the health care systems, as a pharmacological therapy does not yet exist.

Identification of pharmacological targets is further complicated by the heterogeneity of the human population. Individuals with congenital AV abnormalities such as uni- and bicuspid AVs (BAV) typically show earlier onset of FCAVD, attributed to increased mechanical

stress, and manifest clinically relevant determinants of symptomatic AS on average 10 years earlier than patients with the physiological tricuspid AV (TAV)[4]. Furthermore, recent evidence revealed sex-related differences in AS with more fibrotic phenotypes in females as opposed to predominantly calcific remodeling in males[5]. Our analysis homogenized the structural and phenotypical valvular heterogeneity by leveraging the fact that due to its reorganization during FCAVD progression, non- or mildly diseased as well as fibrotic (progressed) and calcific (end-stage) disease stages can be present in the same AV leaflet. Despite major advances in the field, the cardiovascular research community has yet to clarify the precise molecular mechanisms that govern FCAVD onset and progression to facilitate an effective pharmacological intervention.

[1]Faculty of Medicine, Institute for Medical Physics and Biophysics, Leipzig University, Leipzig, Germany. [2]Center of Membrane Biochemistry and Lipid Research, University Hospital and Faculty of Medicine Carl Gustav Carus of TU Dresden, Dresden, Germany. [3]Department of Internal Medicine/Cardiology, Heart Center Leipzig at Leipzig University, Leipzig, Germany. [4]Department of Cardiology, University Medical Center of the Johannes Gutenberg University Mainz and German Center for Cardiovascular Research - Partner Site Rhine-Main, Mainz, Germany. [5]These authors contributed equally: Patricia Prabutzki, Michele Wölk. ✉e-mail: schlottf@uni-mainz.de; maria.fedorova@tu-dresden.de

Major FCAVD hallmarks include differentiation of valve interstitial cells (VICs) to pro-fibrotic and calcific phenotypes, chronic inflammation, immune cell infiltration, and lipoprotein accumulation[6]. Indeed, numerous clinical studies have shown a strong correlation between blood plasma lipid profiles of patients with FCAVD with disease incidence and progression[7–10]. This allowed to identify several lipid-related traits driven by a pathological deposition of low-density lipoprotein (LDL), lipoprotein(a) (Lp(a)) and lipoprotein-associated phospholipase $A_2$ (LpPLA$_2$) mediated conversion of glycerophosphatidylcholine (PC) lipids to the corresponding lyso-forms (LPC), which in turn can be metabolized to bioactive lysophosphatidic acids (LPA) by the action of the secreted phospholipase D, known as autotaxin[11–13]. Similarly, sphingomyelins (SM) delivered by LDL and Lp(a) are converted to the corresponding ceramides (Cer) by the action of sphingomyelinase (SMase), secreted by infiltrated immune cells or resident valve endothelial cells (VEC)[14]. Such metabolic processing of accumulating lipoproteins promotes further lipid deposition, as the generated LPC significantly increases LDL aggregation rates, whereas Cer are positively correlated with both LDL particle fusion and aggregation[15]. However, lipid-lowering therapy with the highly successful anti-atherosclerotic medication class of HMG-CoA reductase inhibitors (statins) failed to show clinical benefits in FCAVD[16–18].

Even though different lipid-related traits of FCAVD progression have been observed, changes in the lipid composition of the affected AV tissues have not been comprehensively characterized. Indeed, qualitative and quantitative descriptions of the specialized lipidomes of human tissues under physiological and pathological conditions have been only sporadically addressed[19–21]. Here, we provide a systematic qualitative and quantitative molecular description of the human AV lipidome covering mildly diseased, fibrotic, and calcific phenotypes. In-depth lipidomics profiling revealed metabolic trajectories associated with the development of fibrosis and calcification, both in TAV and BAV. Importantly, we identified significant sex-specificity in remodeling of AV lipidome upon FCAVD, which was mainly attributed to triacylglycerides (TG) and sphingolipid (SP) metabolism. Finally, all raw and processed lipidomics data produced in this study are made publicly available via Metabolomics Workbench[22] resource, representing the first comprehensive human AV lipidome available for the scientific community for further analysis and data interrogation[23].

## Results

### Reference lipidome of the human AV

To define the molecular composition of the human AV lipidome, we first performed deep lipidomics profiling of pooled samples representing mildly diseased sections of dissected TAVs of patients undergoing surgical aortic valve replacement for severe AS. Mildly diseased TAV regions of 23 patients (13 males and 10 females) with an average age of 72 years were used for lipid extraction (Supplementary Data 1). Reversed-phase chromatography coupled online with mass spectrometry (MS) analysis performed in an untargeted and targeted manner (see Methods section for details) allowed to identify 1073 lipid molecular species (Supplementary Data 2), of which 480 lipids were quantified using class-matching internal standards (Supplementary Data 3 and Fig. 1a). The human TAV lipidome is represented by 28 lipid subclasses spanning over 6 orders of magnitude in lipid concentrations with cholesteryl ester (CE) 18:2 (5.9 nmol/mg wet tissue weight) being the most and SM 32:0;O3 (0.003 pmol/mg) being the least abundant species. The most abundant lipid classes in the human TAV lipidome were CE (total amount 10.80 nmol/mg), followed by TG (1.65 nmol/mg), PC (0.8 nmol/mg), SM (0.76 nmol/mg), and ether glycerophosphatidylethanolamines (etherPE; 0.27 nmol/mg).

High enrichment in neutral lipids (NL) could be attributed to the accumulation of blood lipoproteins in mildly diseased AV sections of elderly individuals. To trace the source of lipid infiltration at the molecular level, we compared the TAV lipid composition with previously published lipidomes of human blood plasma, as a source of circulating lipids[24], and liver, as a major lipoprotein-producing organ[25]. To this end, we employed the uniform manifold approximation and projection (UMAP) algorithm for dimensionality reduction and topological analysis of tissue lipidome similarities. Z-scored lipid abundances and structural features were used as parameters for the projections. UMAP clearly illustrated the close similarity between TAV and plasma lipidomes, whereas liver lipids presented a different topology (Fig. 1b). Specifically, we observed a very close overlap between the TAV and plasma CE with all species sharing similar topology (Supplementary Fig. 1a). TG lipids formed several topological clusters (Supplementary Fig. 1b), with TAV TGs distributed between two clusters—one shared between all three biological samples (mostly TG with 50–54 carbon atoms), and the second formed by TAV and blood plasma TGs. The same was true for PC and LPC lipids (Supplementary Fig. 1c), whereas SP showed highly tissue-specific topologies except for SM, which closely overlapped between TAV and plasma lipidomes (Supplementary Fig. 1d). Taken together, with the exception for Cer lipids, the topological analysis confirmed the close similarity of the TAV and blood plasma lipidomes, especially in respect to the major components of circulating lipoproteins (CE, TG, PC, and SM). Furthermore, when compared in terms of total CE to TG ratio (Fig. 1c), the TAV lipidome of mildly diseased sections was more similar to blood plasma[24], than other NL-rich tissues, including liver[25] or adipose tissue[24].

Next, we examined whether lipid molecular signatures can help to define which circulating lipoproteins contribute to the lipidome of mildly diseased TAV. Based on immunohistochemistry and proteomics data, a major contribution of both LDL and Lp(a) in TAV lipid deposition was proposed[26,27]. Indeed, based on the CE to TG ratio a more significant impact of CE-rich rather than TG-rich lipoproteins was confirmed (Fig. 1c). Additionally, comparison of the SM to PC ratio in TAV versus LDL, Lp(a), very low-density (VLDL) and high-density (HDL) lipoproteins for which different relative contributions of SM and PC lipids were reported[28–33] revealed its close similarity to LDL and Lp(a) (Fig. 1d). This comparative analysis implies that a substantial fraction of the TAV lipidome, especially with respect to NL and SM, derives from retained and accumulated LDL and Lp(a) particles in mildly diseased TAV regions of elderly individuals. This is further supported by the fact that CE 18:2, a marker of extracellular LDL accumulation[19], is by far the most abundant lipid in mildly diseased TAV (Fig. 1a).

Interestingly, the lipidome of mildly diseased TAV revealed a remarkable richness and diversity of SP lipids when compared to blood plasma. In addition to highly abundant SM sharing plasma SM topology, various subclasses of Cer formed TAV-specific clusters (Supplementary Fig. 1d). It is particularly interesting since these lipids recently gained high significance as potential markers for cardiovascular diseases (CVD)[34]. TAV Cer showed very high structural diversity, both in terms of sphingoid bases and fatty acyl (FA) chains (Fig. 1e), with sphingosine (18:1;O2) being the most abundant base (74% of all Cer), followed by deoxysphingosine (18:1;O; 9.8%) and sphingodienine (18:2;O2; 6.7%), whereas the most abundant acyl chains were represented by FA 24:0 (31.3%), FA 24:1 (21.1%), and FA 16:0 (18.3%).

Taken together, using deep lipidomics profiling, we quantitatively described the lipidome of mildly diseased human TAV of patients with FCAVD. By comparing the TAV lipidome with previously published lipidomes of other human tissues (adipose tissue, liver, and blood plasma), we could demonstrate a strong similarity between TAV and plasma lipid compositions. Furthermore, using the CE/TG and SM/PC ratio as a proxy for different lipoproteins, enrichment of LDL and Lp(a) particles over VLDL and HDL was confirmed. Interestingly, in addition to the expected accumulation of typical lipoprotein-derived lipids (CE, TG, PC, and SM), we could demonstrate a large diversity of ceramides as a specific molecular feature of the human TAV lipidome.

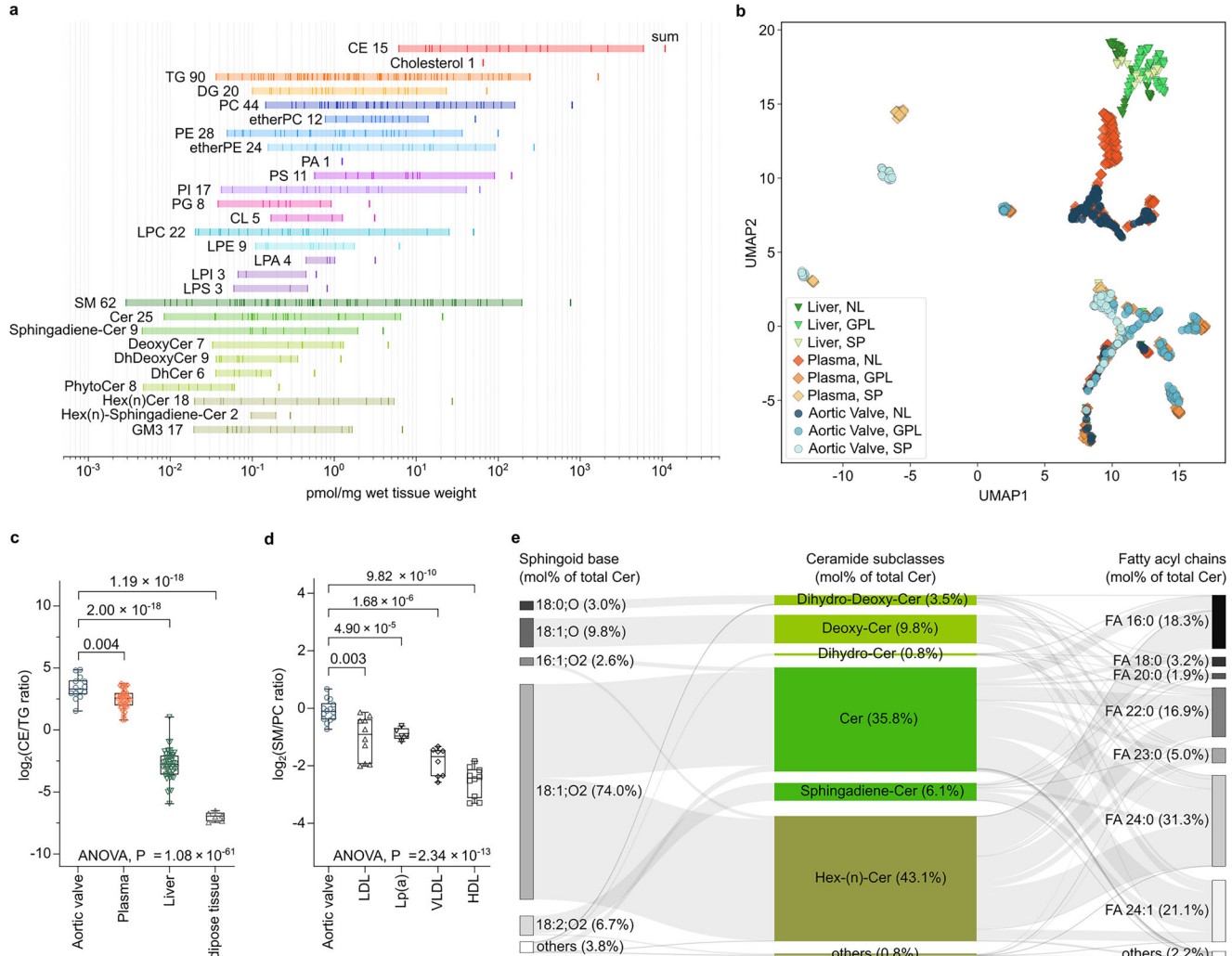

**Fig. 1 | Tissue-specific and global lipid topologies of the human aortic valve lipidome. a** Quantitative distribution of lipid subclasses and corresponding molecular species in mildly diseased human TAVs of elderly individuals. Total concentration for each lipid subclass is shown by the bold line (sum), and the concentrations of each lipid molecular species are marked by thin lines within the shaded region. **b** Topological comparison using uniform manifold approximation and projection (UMAP) of human TAV, blood plasma, and liver lipidomes. **c** Total CE to TG ratio in TAV ($n = 14$), blood plasma ($n = 50$), liver ($n = 49$), and adipose tissue ($n = 6$) lipidomes. **d**, Total SM to PC ratio in TAV ($n = 14$) and different lipoprotein fractions ($n = 5$). *P* values were calculated by Welch's T-test or one-way ANOVA across all groups. Boxplot elements represent centerline, mean; box limits, 25% and 75% quartiles; vertical lines connect minimum and maximum values. Dots represent independent samples. **e** Sankey plot illustrating the relative abundance of TAV ceramide (Cer) subclasses (central), composed of the corresponding sphingoid bases and fatty acyl chains shown on the left and right sides, respectively. Source data are provided as a Source Data file.

## Lipid traits of FCAVD progression are driven by lipoproteins infiltration and metabolic conversion

Next, we focused on identifying the molecular signatures of the TAV lipidome in the advanced pathological stages of FCAVD. To this end, the AV of patients undergoing valve replacement for severe AS was dissected into different regions, namely mildly diseased, fibrotic, and calcific sections (Fig. 2a). Relative to the mildly diseased sections, total lipid content increases in fibrotic (2.3-fold) and calcific (2-fold) tissue sections. CE, cholesterol, and SPs contributed the most to the elevated lipid load. Thus, CE concentrations in fibrotic and calcific sections were increased 2.7- and 2.3-fold, respectively (Fig. 2b). Free cholesterol was significantly elevated in the fibrotic (2.6-fold) and even further increased (3.0-fold) in the calcific stage (Fig. 2c). Interestingly, total TG load did not change much between different FCAVD progression stages and in general showed high inter-individual variability (Fig. 2c). Similarly, no increase in the load of glycerophospholipids was observed, whereas SPs showed large accumulation in fibrotic (2.0-fold) and calcific (1.9-fold) sections of TAV (Fig. 2d). Quantitative comparison of all lipid subclasses

between mildly diseased, fibrotic and calcific TAV sections is provided in Supplementary Fig. 2.

To investigate the TAV pathophysiological lipidome remodeling further, we focused on the known lipid-related traits of FCAVD progression, namely (I) the LDL – LpPLA₂ – LPC – LPA axis and (II) SMase-induced Cer accumulation. As expected, validating our approach, we observed a significant increase in the absolute LPC concentrations (Supplementary Data 3) as well as the ratio of total LPC to PC lipids in the fibrotic (2.8-fold) and calcific (4.0-fold) TAV relative to the mildly diseased sections (Fig. 2e), indicating high LpPLA₂ activity promoting consequent lipoprotein aggregation and deposition. Similar trends (1.7- and 1.8-fold increase in fibrotic and calcific sections, respectively) were observed regarding the LPE to PE ratio (Supplementary Fig. 3).

Several reports suggested that the fibro-calcific response in FCAVD is not driven by LPC itself, but rather by LPC-derived LPA produced by the phospholipase D autotaxin[12]. Despite previously published high LPA concentrations in AV tissue[21,35], we were unable to detect any endogenous LPA or PA species in lipid extracts obtained by conventional methods (i.e., Folch extraction; Supplementary Fig. 4).

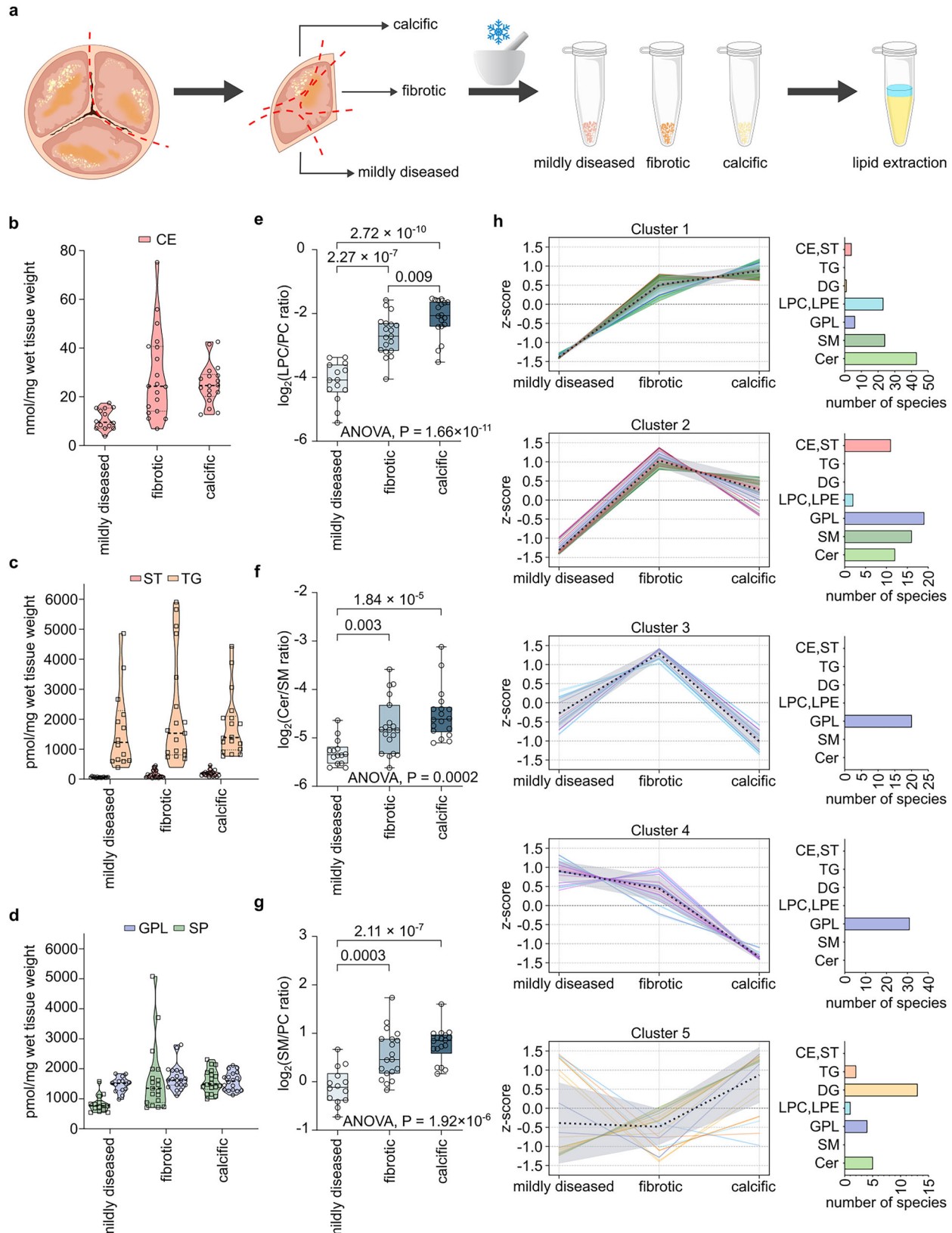

Considering the proposed relevance of LPA in FCAVD pathology, we therefore employed acidified methanol-chloroform based extraction[36], followed by targeted LC-MS/MS analysis in order to analyze PA and LPA specifically (Supplementary Fig. 4). To note, PA and LPA analysis requires careful analytical optimization, as both extraction and MS analysis in complex biological matrices can result in

artificial PA and LPA generation by either acid-mediated hydrolysis or in-source fragmentation of other GPL[37–39]. Considering all the analytical limitations mentioned above, we detected and quantified one PA and four LPA molecular species, of which only LPA 16:0 showed a significant difference and was slightly elevated in calcific TAV sections (Supplementary Fig. 4 and Supplementary Data 3).

**Fig. 2 | Overall lipid accumulation and differential metabolic trajectories driving aortic valve fibrosis and calcification. a** Schematic representation of AV tissue dissection into mildly diseased, fibrotic, and calcific sections for a separate lipid extraction. Violin plots showing the mean total concentration of cholesteryl esters (CE) (**b**), triacylglycerides (TG) and free cholesterol (ST) (**c**), glycerophospholipids (GPL) and sphingolipids (SP) (**d**) in mildly diseased, fibrotic and calcific tissue sections. Violin plot elements represent dashed centerline, median; dotted lines, 25% and 75% quartiles. Dots represent biologically independent samples (*n* = 21). Total LPC to PC (**e**), Cer to SM (**f**), and SM to PC (**g**) ratios in mildly diseased, fibrotic, and calcific TAV sections. *P* values were calculated by two-sided *t* test or one-way ANOVA across all groups. Box plot elements represent centerline, mean; box limits, 25% and 75% quartiles; vertical lines connect minimum and maximum values. Dots represent biologically independent samples (*n* = 21). Color-coding indicates pathophysiological stage: light blue–mildly diseased; blue–fibrotic; dark blue–calcific. **h** Lipid Trend Analysis via Gaussian mixture model clustering (number of clusters = 5) of z-scored concentrations for 237 significantly regulated lipids (determined by one-way ANOVA $P < 0.05$, *n* = 21; *P* values are provided in Supplementary Data 4a) along FCAVD progression in TAV. Black dashed line presents mean values of each phase. The grey area presents the mean values ∓ standard deviation area. Line color indicated different lipid classes: red–cholesterol and CE; orange–TG; yellow–DG; LPC and LPE–light blue; GPL–dark blue; SM–dark green; Cer–light green. CE cholesteryl esters, Cer ceramides; DG diacylglycerides; GPL glycerophospholipids; LPC lysophosphatidylcholines; LPE lysophosphatidylethanolamines; PC phosphatidylcholines; SM sphingomyelins; ST free cholesterol; TG triacylglycerides. Source data are provided as a Source Data file.

Next, we investigated the ratio between Cer and SM lipids as a second prominent trait associated with increased lipoprotein deposition[15]. Not only was the total SM content 1.9- and 1.8-fold higher in fibrotic and calcific TAV relative to mildly diseased sections, but also the Cer/SM ratio rose 1.5- and 1.8-fold, respectively, with FCAVD progression. This indicates a high conversion rate of infiltrating SM to the corresponding Cer by the action of SMase, priming lipoprotein aggregation and deposition (Fig. 2f). Generally, LDL aggregation was shown to correlate positively with SM and Cer lipids and negatively with PC lipids[32]. We could further confirm this lipid-related FCAVD trait and show that the total SM to PC ratio steadily increased with disease progression corresponding to 1.6- and 1.8-fold higher values in fibrotic and calcific sections relative to mildly diseased TAV (Fig. 2g). Thus, by defining mildly diseased, fibrotic and calcific TAV lipidomes, we could confirm, in a quantitative manner, that the major traits in FCAVD pathogenesis are associated with increased LDL and Lp(a) aggregation and deposition, namely PC to LPC and SM to Cer conversions, serving as a proxy for LpPLA$_2$ and SMase activities.

## Lipidomics reveals extrinsic and intrinsic signatures of aortic valve fibrosis and calcification

Lipid trends described above represent an important part of the TAV lipidome remodelling, driven mainly by lipoprotein aggregation and deposition. However, during FCAVD pathogenesis, the TAV lipidome undergoes alterations induced by intrinsic remodelling, driven by VIC transition to myofibroblastic and osteoblastic phenotypes, VEC endothelial to mesenchymal transition, and immune cell infiltration as well. To dissect extrinsic and intrinsic lipidomic signatures of the TAV pathology, we performed Lipid Trends Analysis, utilizing Gaussian mixture model clustering of 237 lipid species significantly regulated between mildly diseased, fibrotic, and calcific TAV sections (ANOVA $P < 0.05$; Fig. 2h; Supplementary Data 4a).

The first two clusters can be attributed to the extrinsic trends mediated by lipoprotein infiltration, characterized by progressive (cluster 1) or fibrotic stage tipping (cluster 2) lipid deposition. Clusters 1 and 2 are formed by 101 and 60 lipids, respectively, including major lipid species likely of lipoprotein origin, namely cholesterol, CE, and SM. Interestingly, most of the CE belonged to cluster 2 showing that deposition of lipoproteins occurs mainly in the fibrotic stage. Further, the total load of CE lipids does not increase during the progression from the fibrotic to the calcific stage. On the other hand, cluster 1, in addition to the infiltrated lipids (cholesterol and SM), also included LPC, LPE, and Cer, produced locally from lipoprotein-derived lipids by the action of LpLPA$_2$ and SMase. These lipids showed progressive accumulation over the pathological stages of FCAVD, with steep elevation already at the fibrotic stage and a moderate increase in calcific TAV. Thus, clusters 1 and 2 can be assigned as "lipoprotein infiltration" and "lipoprotein processing" trends, respectively.

Clusters 3 and 4 showed not only different trends but also very different compositions compared to the lipoprotein-associated clusters 1 and 2, and thus, we propose that they might represent TAV intrinsic lipidome remodeling. Indeed, neither CE nor SM species are present in these clusters, which are solely represented by various GPL lipids (Fig. 2h). Abundances of lipids from cluster 3 tip at the fibrotic stage and rapidly decline with TAV calcification, possibly indicating the increase in TAV cell populations due to VIC and VEC transformations and immune cell influx along with fibrosis development. Indeed, cluster 3 contains 20 lipids, of which 18 are represented by major membrane building PC, PE, and etherPE species. Cluster 4, on the other hand, shows a progressive decline in the abundances of 31 lipid species, especially evident for calcific TAV. In addition to a few PC, PE, and polyunsaturated fatty acyl (PUFA)-rich etherPE lipids, cluster 4 includes cardiolipins (CL; four out of five detected species) and phosphatidylserine (PS; 10 out of 11 detected species) lipids. We attributed cluster 4 to major tissue degenerative events occurring in calcific TAV, accompanied by the massive loss of PUFA etherPE due to increased oxidative stress and cell death, evidenced by a sharp decline of mitochondria-specific CL lipids. PS lipids depletion in calcific TAV represents an interesting phenomenon and serves as a specific marker of progressive tissue calcification. Finally, the remaining cluster 5 combines a set of mixed lipid species (25 lipids) with no common trend.

Taken together, using lipid trend analysis, we dissected extrinsic (associated with lipoprotein infiltration) and intrinsic (driven by TAV cell populations) metabolic trajectories of lipidome remodeling in FCAVD progression. Thus, we demonstrated that lipoprotein-derived lipids undergo pathological processing in TAV, leading to the accumulation of lysoGPL and Cer, mediated by the activity of LpPLA$_2$ and SMase, respectively. Among intrinsic pathways, lipidome signatures indicated increased cell density at the fibrotic stage, followed by tissue degeneration in calcific TAV. Interestingly, one of the most prominent hallmarks of the calcific TAV lipidome was a massive reduction in PS lipids, which are probably involved in the calcification process due to their anionic nature and, thus, may serve as markers of FCAVD progression.

## Lipidome remodeling during FCAVD progression is sex-specific

Multiple lines of evidence indicate that biological sex plays an important role in FCAVD progression, as with the same hemodynamically defined degree of AS, women develop a more fibrotic phenotype, while men are characterized by a higher extent of calcification[40–42]. Indeed, AVs from female patients included in this study generally presented with more of a fibrotic phenotype in comparison to samples of male patients, which showed higher calcification (Fig. 3a). Thus, we compared the differences in lipidome signatures of AV sections for male and female patients.

First, striking differences in the extent of the total lipid deposition between male and female TAVs were detected (Fig. 3b-d). Mildly diseased TAV tissues of males had a higher lipid load (16.5 vs 12.6 nmol/mg wet tissue weight in male and female TAV, respectively), mainly due to the elevated content of total CE and TG lipids. However, upon development of fibrosis and calcification, female TAV sections

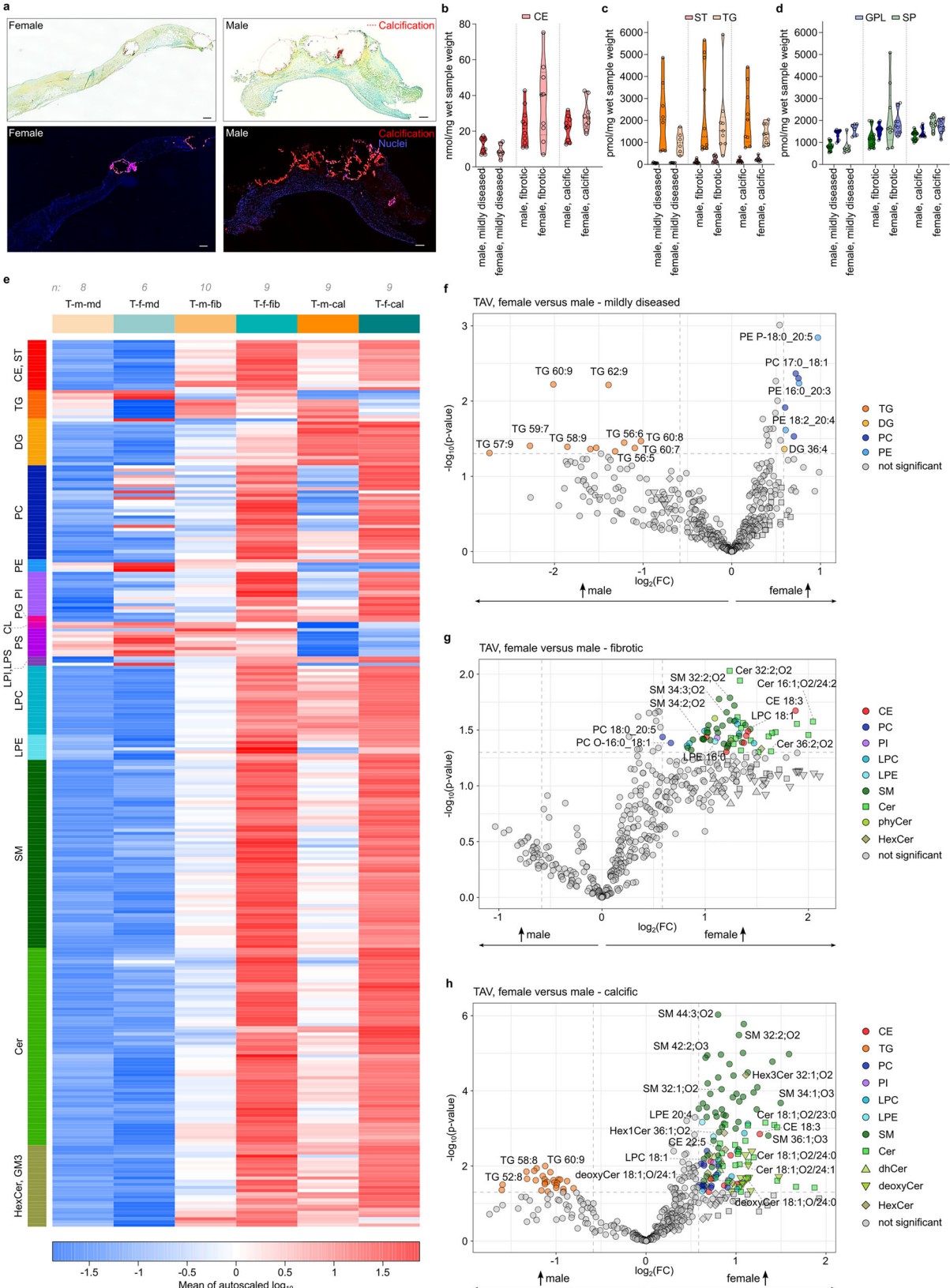

accumulated higher amounts of lipids. Thus, relative to mildly diseased sections, male TAV increased their lipid load up to 1.7- and 1.6-fold, whereas female patients showed a much higher lipid deposition with a 3.4- and 2.7-fold increase in fibrotic and calcific stages, respectively. These differences occurred mostly during the fibrotic stage and were mainly driven by the 4-fold increase in CE lipids in female TAVs

(Fig. 3b). Overall, all lipid classes, except GPL, showed higher fold changes in both fibrotic and calcific stages of female patients (Fig. 3c, d) when compared to male TAV sections.

To take a deeper look into sex-specific FCAVD lipid phenotypes, we tracked lipidome-remodeling at different pathological stages in a sex-specific manner (Fig. 3e and Supplementary Fig. 5). In total, 283

**Fig. 3 | Fibro-calcific remodeling of aortic valve lipidome is sex-specific.**
**a** Extracellular matrix remodeling in female and male FCAVD TAV sections visualized with Movat's pentachrome staining (top panels) shows collagen (yellow) and glycosaminoglycans (turquoise), and OsteoSense and DAPI staining (bottom panels) for calcific mineralization (red) and nuclei (blue), respectively. Bar corresponds to 100 μm. Violin plots showing the mean total concentration of cholesteryl esters (CE) (**b**) triacylglycerides (TG) and free cholesterol (ST) (**c**) glycerophospholipids (GPL) and sphingolipids (SP) (**d**) in mildly diseased, fibrotic, and calcific TAV tissue sections of male and female patients, respectively. Violin plot elements represent dashed centerline, median; dotted lines, 25 and 75% quartiles. Dots represent biologically independent samples ($n = 21$). **e** Heatmap illustrating normalized abundances of the significantly regulated lipids (one-way ANOVA p < 0.05) in TAV lipidomes (T) of female (f) and male (m) individuals ($n = 21$) across different pathophysiological stages of FCAVD (md−mildly diseased; fib−fibrotic; cal−calcific). Volcano plots illustrating lipid species significantly regulated between female versus male individuals ($n = 21$) in mildly diseased (**f**), fibrotic (**g**), and calcific (**h**) TAV tissue sections determined by fold change (FC > 1.5) analysis and two-sided $t$ test ($P < 0.05$). Source data are provided as a Source Data file.

lipids showed significant differences (ANOVA $P < 0.05$; Supplementary Data 4b) between mildly diseased, fibrotic, and calcific TAV of male versus female individuals. Although many lipid alterations represented the major FCAVD development trends discussed above, female patients showed much higher accumulation of lipoprotein-derived lipids (CE, PC, SM) as well as the products of their metabolic processing (LPC, Cer) along the FCAVD progression. The major sex-specific differences between the pathological stages seem to be driven mainly by CE and SP in females and TG lipids in males. Indeed, TG lipids did not show any significant differences in FCAVD progression when male and female samples were analyzed together, and emerged as one of the main sex-specific trends already in the mildly diseased stage.

Next, we performed a pairwise comparison of female vs male TAV lipidomes at each pathological stage (Fig. 3f–h; Supplementary Data 5a). Male mildly diseased TAVs contained higher amounts of TG lipids carrying PUFA acyl chains (C56-C62, with 5–9 double bounds), whereas female TAVs had higher levels of PUFA containing plasmalogen PE (P-PE) and PC carrying mostly saturated and monounsaturated acyl chains (MUFA) (Fig. 3f). However, the major differences became apparent when comparing fibrotic and calcific sections, in which 66 and 161 lipids were differentially regulated in female vs male fibrotic and calcific TAV sections, respectively (Fig. 3g, h). In fibrotic TAV sections, all significantly differentially abundant lipids were higher in female patients and included CE, SM, Cer, and lysoGPL lipids, indicating a higher pathological conversion (PC to LPC and SM to Cer) of infiltrating lipids, which in turn results in a higher lipoprotein aggregation and deposition.

This sex-driven dimorphism in lipid accumulation was further evident in the calcific stage (Fig. 3h). Here, 134 and 27 lipids were significantly increased in abundance in female or male calcific TAVs, respectively. The 27 lipids, significantly upregulated in male TAVs, were all TG lipids, among which PUFA-rich species dominated again. Interestingly, at the quantitative level, the overall TG load in calcific male TAV did not increase relative to mildly diseased sections (roughly 2.2 nmol/mg wet tissue weight in both mildly diseased and calcific male TAV), whereas female TAVs increased their TG load 1.4-fold (1.0 vs 1.4 nmol/mg wet tissue weight in mildly diseased and calcific female TAV, respectively) (Fig. 3c). Thus, significantly higher concentrations of TG in calcific male TAVs indicate sex-specific differences at the level of particular molecular species, namely PUFA-rich TG. Similar to the fibrotic stage, major lipid classes upregulated in calcific female TAV are represented by CE, lysoGPL, and SP lipids (Fig. 3h).

Almost all measured SP subclasses were upregulated in female calcific TAV, including SM and regular Cer, as well as dhCer, deoxyCer, sphingadienine-Cer, phytoCer, and glycosylated species (Fig. 3h). While SM were represented by a diverse composition of molecular species ranging from SM 28:1;O2 to SM 54:2;O3, accumulated Cer showed distinct acyl chains enrichment. Independent of the sphingoid base, the vast majority of Cer lipids in female calcific TAV carried very long chain fatty acids (VLCFA) ranging from C22 to C26. Thus, we propose that the SP pool in calcific TAVs of female patients may be formed, not only by the conversions of lipoprotein-derived SM to the corresponding Cer, but also via AV-local SP metabolism. In favor of this hypothesis, we could detect sex-specific upregulation of dhCer, which

are precursors of Cer lipids in de novo synthesis pathways, carrying the same VLCFA acyl residues as elevated Cer species.

Taken together, we demonstrated significant sexual dimorphism in the lipidome signatures of FCAVD. Mildly diseased TAV sections of male patients had higher lipid content due to the elevated CE and TG lipids. However, female TAVs accumulated much more lipids with the development of fibrosis, mainly due to the massive deposition of lipoprotein-derived lipids and their metabolic conversion. TG and SP lipids emerged as a main discriminator of sex-specific lipidome remodeling in FCAVD, with female individuals accumulating significantly higher levels of SM and Cer lipids.

### Lipidome signatures discriminate tricuspid and bicuspid valve morphology

Finally, we compared lipidome signatures of TAV vs BAV. Pairwise comparison between pathological stages of BAV and TAV revealed that mildly diseased sections of BAV contained more SP lipids represented specifically by VLCFA species of a variety of SP subclasses (phytoCer, Cer, sphingadienine-Cer, SM, GM3, and Hex2Cer) (Fig. 4a; Supplementary Data 5b). Fibrotic and calcific BAVs showed lower contents of TG lipids (Fig. 4b, c). Interestingly, in calcific BAVs, downregulated lipids were PUFA-rich TG as well as VLCFA deoxyCer.

Contrary to the previous observations in TAV, heatmap representation of significantly regulated lipids (361 lipids; ANOVA $p \leq 0.05$; Supplementary Data 4c) showed that BAV displays less sex-driven dimorphism of disease progression. Accumulation of lipoprotein-derived lipids (CE, PC, SM) and products of their metabolic processing (LPC, Cer) showed quite similar trajectories in male and female BAVs (Fig. 4d and Supplementary Fig. 6). Whereas, female TAVs showed higher levels of SP, and male TAV had higher content of TG relative to BAVs at all disease stages. It is important to note that individuals with BAVs typically experience an earlier onset of FCAVD, with clinically manifested symptomatic AS occurring on average 10 years earlier than in patients with the physiological TAV. Subsequently, the observed variability in lipidomes between TAV and BAV may be attributed to differences in the age of patients with bicuspid versus tricuspid AVs.

Taken together, fibrotic and calcific TAV and BAV lipidomes share large similarities. However, BAV lipid signatures of FCAVD are not sex-specific and overall close to the pathological trajectories detected in female TAVs. However, the specific enrichment of VLCFA SP lipids in mildly diseased BAV sections indicates that these lipid species might be an early marker of pathological predisposition.

### Discussion

Lipid dysmetabolism is a well-known driver of FCAVD onset and progression, however, limited insight into the underlying mechanisms prevents further development of pharmaceutical interventions. Indeed, despite certain similarities with atherosclerosis in terms of an elevated tissue deposition of CE-rich lipoproteins, interventions based on the inhibition of the cholesterol biosynthesis pathway with statins did not show a significant reduction in AS progression or severity[43].

To support the mechanistic understanding of dyslipidemia in fibro-calcific remodeling, we performed in-depth quantitative lipidomic characterization of human AVs along FCAVD progression,

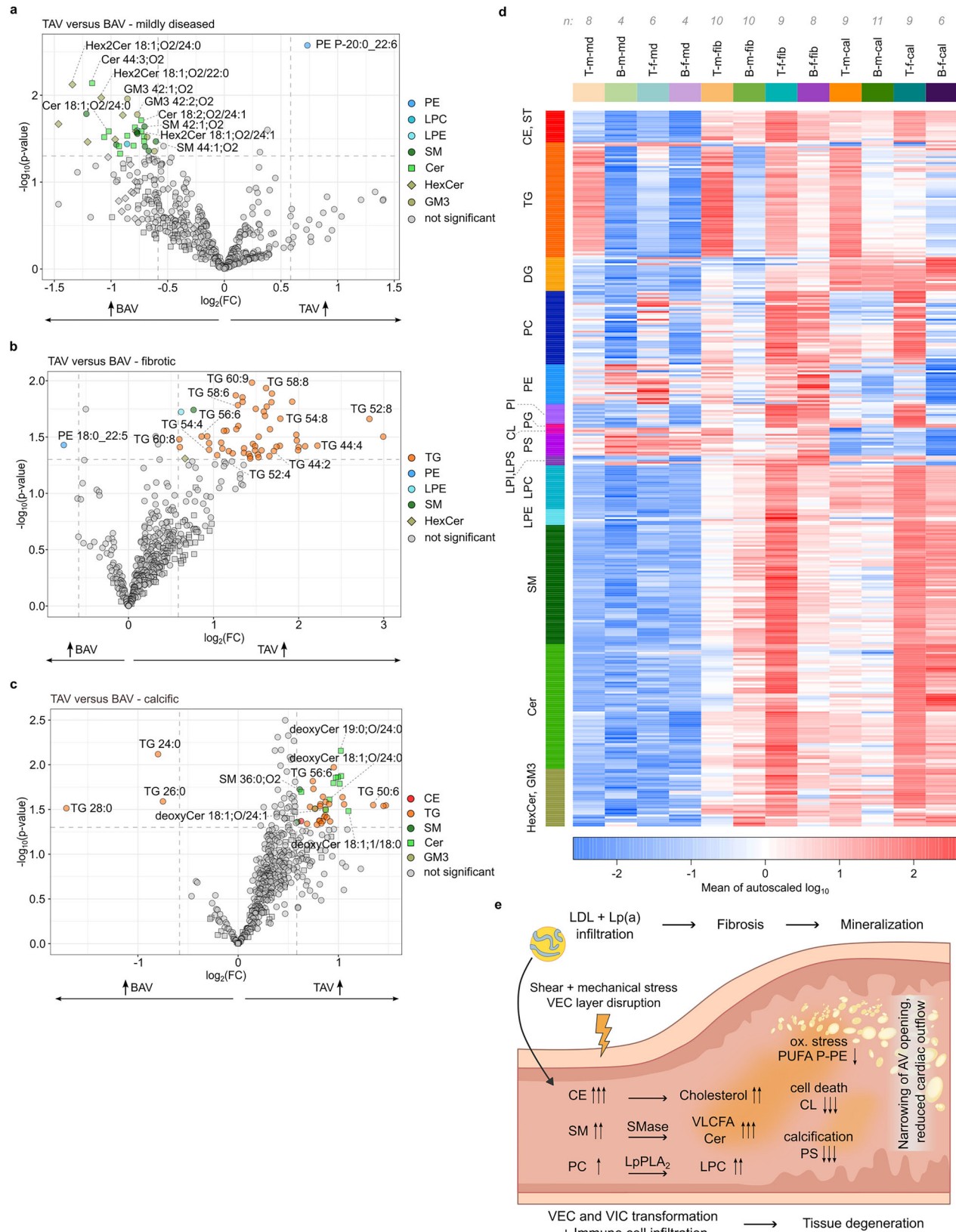

**Fig. 4 | Lipidomics signatures discriminate tricuspid (TAV) from bicuspid (BAV) AV in FCAVD.** Volcano plots illustrating lipid species significantly regulated between TAV versus BAV lipidomes of mildly diseased (**a**), fibrotic (**b**), and calcific (**c**) AV tissue sections determined by fold change (FC > 1.5) analysis and two-sided *t* test (*P* < 0.05, *n* = 41). **d** Heatmap illustrating normalized abundances of significance regulated lipids (one-way ANOVA *p* < 0.05) of TAV (T) and BAV (B) lipidomes of female (f) and male (m) individuals (*n* = 41) across different pathophysiological stages of FCAVD (md–mildly diseased; fib–fibrotic; cal–calcific). **e** FCAVD progression is characterized by the same lipidome alterations in TAV and BAV. Source data are provided as a Source Data file.

covering mildly diseased, fibrotic, and calcific stages. Over 1070 lipid molecular species were annotated, of which 480 were quantified. Using UMAP-based topological analysis of selected human lipidomes, we could demonstrate that the AV lipidome of mildly diseased AVs closely overlaps with the lipidome of blood plasma in terms of main lipoprotein-derived lipids (CE, TG, SM, PC), whereas Cer showed an AV-specific pattern. Lipoprotein-related molecular signatures confirmed AV enrichment with lipids from CE-rich LDL and Lp(a), complementing previously published immunohistochemistry and proteomics data[26,44].

By mapping lipidomic signatures to the different stages of FCAVD, both extrinsic and intrinsic metabolic trajectories were identified (Fig. 4e). Lipoprotein accumulation, already evident in mildly diseased sections, peaked at the fibrotic stage and was characterized by a significant buildup of CE, free cholesterol and SP lipids, with TG having only minor impact. Importantly, the infiltration of lipoprotein-derived lipids led not just to a passive lipid deposition but was accompanied by the active metabolic processing of PC to LPC and SM to Cer by the action of LpPLA$_2$ and SMase, respectively. Elevated LPC generally reflected the composition of the infiltrated PC precursors. On the other hand, although diverse SM species accumulated in diseased AVs, elevated Cer were mainly represented by species carrying VLCFA chains. This apparent specificity is not entirely clear and might be attributed to SP metabolism within AV cell populations in addition to the SMase action on lipoprotein-derived SM. Interestingly, the enzyme responsible for the synthesis of VLCFA Cer, ceramide synthase 2 (CerS2), is known to be highly expressed in the heart[45].

Considering the well-established correlation between the auto-taxin activity with the severity of FCAVD[12], endogenous levels of LPA lipids, which are often named as bioactive mediators driving FCAVD progression[46], were evaluated. However, detection of endogenous LPA in AV samples processed with a conventional lipid extraction protocol was not possible. Using tailored extraction and targeted MS detection, we could quantify four LPA species of which only LPA 16:0 showed a slight but significant elevation in calcific AV. Thus, we propose that LPA generation and biological activities may be rather transient and challenging to detect in advanced fibrotic and calcific lesions.

Active metabolic conversion of lipoprotein-derived PC and SM to the corresponding LPC and Cer increases lipoprotein aggregation, fusion, and retention in AV, representing a vicious cycle escalating lipid deposition and AV steatosis. Indeed, Cer lipids attracted much attention in cardiovascular research. Circulating levels of four particular Cer molecular species have been used as a prognostic scoring test named the Coronary Event Risk Test (CERT)[47,48], which showed to be superior in predicting cardiovascular risk compared to traditional markers such as LDL-cholesterol (LDL-C)[49,50]. Considering the correlation between CVD severity and Cer-based CERT score, it is possible to speculate that Cer accumulated in AV or atherosclerotic lesions can leak back into the circulation, reflecting the valvular and vasculature status.

Interestingly, it was demonstrated that Cer composition in liver, the main site of SP synthesis and lipoprotein secreting organ, is positively correlated not only with the corresponding LDL SM species, but also with LDL aggregation[51]. Overall, the composition of hepatic TG, SP, and PC lipids is closely related to that of LDL[51]. Thus, Cer synthesized in the liver is released into circulation in the form of the corresponding SM within lipoproteins, which, when trapped in the AV, are converted back to the bioactive Cer species[52]. Thus, one can propose a direct link between liver dysmetabolism and FCAVD via SP lipid signatures. Indeed, correlations between AS and alcoholic and non-alcoholic fatty liver diseases were demonstrated[53–55]. Interestingly, hepatocyte-specific inhibition of CerS2 reduced levels of circulating Cer used in the CERT score[56], further underlying the significance of SP lipids in the organ-to-organ metabolic cross-talk.

In addition to FCAVD metabolic trajectories driven by lipoprotein infiltration, we identified AV-intrinsic lipidomic signatures. Accumulation of major membrane-building GPL at the fibrotic stage probably

reflects increased cell populations due to the transformation and proliferation of AV resident cells and/or infiltrating immune cells. The calcific TAV lipidome showed degenerative signatures characterized by a massive loss of PUFA etherPE, most probably due to their oxidative degradation, and mitochondria-specific CL lipids indicating extensive cell death. Interestingly, we observed a sharp decline in the levels of PS lipids as a hallmark of progressive AV calcification. PS are one of the major anionic lipids enriched at the inner leaflet of plasma membranes and the cytosolic side of late endocytic compartments[57]. Thus, extensive PS depletion might indicate both disorganization of plasma membranes and/or cellular secretory and endocytic pathways. PS were shown as crucial players in the intracellular trafficking of LDL-derived cholesterol from the plasma membrane to the ER[58]. In conditions of massive cholesterol overload, PS homeostasis might be dysregulated. However, the most plausible explanation is probably the role of anionic PS lipids in calcification, where negatively charged head groups of PS lipids serve as hydroxyapatite nucleator sites[59].

Increasing evidence indicates that biological sex plays an important role in FCAVD[40–42]. Thus, one of the major findings of this study is that metabolic trajectories in FCAVD progression are highly sex-specific. Major sex-specific traits are defined by higher content of PUFA TG in male AV and elevated accumulation of SM and Cer lipids in female tissues. These sex-specific differences in the FCAVD lipidome might reflect the differences in blood lipidomes of male and female individuals[60–62]. Interestingly, in FCAVD, women exhibit a more fibrotic FCAVD phenotype, which can be associated with higher levels of Cer in female AVs, as identified here. SP in general and Cer in particular are emerging mediators of tissue fibrosis. The link between SP metabolism and fibrosis onset and development was shown for liver[63], kidney[64], cardiac[65], and cystic fibrosis[66]. Among signaling pathways classically attributed to fibrosis development, TGF-β signaling was shown to be directly correlated with Cer levels[67]. Interestingly, TGF-β signaling is upregulated in women relative to men with AS[68,69]. Thus, based on our lipidomics data, we propose a possible mechanistic link between sex-specific TGF-β signaling, Cer accumulation, and fibrosis severity, which might open a window of opportunities for therapeutic intervention to limit fibrotic AV remodeling. Interestingly, metabolic trajectories of BAV along FCAVD development were less sex-specific and generally more similar to the female TAV, thus associated with higher SP accumulation, possibly reflecting a prevalence of the fibrotic stage. It can also be interpreted that the hemodynamic effects are more detrimental for the pathological process in BAV, rather than the sex-specific mechanisms that may be more relevant in TAV. Further, age differences between TAV and BAV FCAVD patients might be responsible for some of the observed differences in AV lipidomes.

In conclusion, this study identified distinct lipidomic signatures in FCAVD and provides a molecular lipid map of the AV, highlighting both lipoprotein-derived and valve-intrinsic lipid alterations. The identification of Cer accumulation as a key factor in fibrotic FCAVD progression, which represents the dominant phenotype in female FCAVD, opens avenues for the development of pharmacological interventions aimed at reducing Cer levels, either through inhibition of their biosynthesis or enhancement of their degradation[34,70,71]. Thus, targeting SP rather than cholesterol metabolism might be a promising option for FCAVD treatment. Importantly, we identified significant sex-specific responses in FCAVD progression. With women presenting significantly higher levels of AV Cer, future development in SP-based diagnostics and therapeutic interventions should be performed in a sex-specific manner. Indeed, the general misperception that women are less susceptible to CVD stems from an under-representation of women in clinical trials. Even in cohorts used for the validation of CERT or sphingolipid-inclusive score, as strong CVD event predictors, 72% and 77% of the cohorts, respectively, were represented by male individuals[72,73]. The sex-specific differences in

FCAVD identified in this study call for greater inclusion of women in research and clinical trials to improve diagnostic accuracy and develop tailored therapeutic strategies that address these disparities. Furthermore, the study reveals the interplay between systemic lipid metabolism and valvular lipid homeostasis, suggesting that circulating sphingolipids may play a role in FCAVD pathogenesis. These insights strengthen the rationale for investigating organ-to-organ metabolic cross-talk in FCAVD.

The obvious limitation of our study is the small sample size, consisting of 55 patients, which would call for further validation. To this end, all lipidomics data obtained here are made publicly available to the community both as raw MS files and processed datasets (ST003364 at Metabolomics Workbench), to facilitate further data interrogation and, importantly, to design targeted MS assays tailored to the human AV lipidome for high-throughput validation in larger cohorts of samples[23].

## Methods

### Chemicals
Acetonitrile (ULC/MS-CC/SFC grade, >99.97%), methanol (UHPLC-MS grade, >99.97%), isopropanol (ULC/MS-CC/SFC grade, >9.95%), and formic acid (ULC/MS-CC/SFC grade, >99%) were obtained from Biosolve B.V. (Valkenswaald, Netherlands). Hydrochloric acid was purchased from Merck KGaA (Darmstadt, Germany). Acetic acid, ammonium formate (MS grade), butylated hydroxytoluene (BHT), chloroform (analytical grade, EMSURE®), diethyl ether, hexane, primuline, and triethylamine were purchased from Sigma Aldrich Chemie GmbH (Taufkirchen, Germany). SPLASH®LIPIDOMIX®, Cer/Sph Mixture I, Cer 18:0;O2/12:0, Cer 18:0;O3/8:0, Cer 18:1;O2/17:0;O, CL 18:2/18:2/18:2/18:2 (d5), LPA 16:0, LPA 17:1, LPA 18:0, LPA 18:1, LPA 20:4, PA 17:0/17:0, PA 16:0/18:1 and TLC standards, including TG 18:0/18:0/18:0, FA 18:0, Cholesterol, DG 16:0/16:0/0:0, DG 16:0/0:0/16:0, MG 18:0/0:0/0:0, PG 16:0/18:1, PE 16:0/18:1, PI 16:0/18:1, PS 16:0/18:1, LPE 18:1, PC 16:0/18:1, SM 18:1;O2/16:0 and LPC 18:1, were obtained from Avanti Polar Lipids Inc (Alabaster, USA). Cer 18:0;O2/8:0 was from Sigma Aldrich. Water (resistance $R > 18 \, M\Omega/cm$, total organic content less than 10 ppb) was purified on a PureLab Ultra Analytic system (ELGA Lab water, Celle, Germany) or obtained from Biosolve B.V. (ULC/MS-CC/SFC grade, Valkenswaald, Netherlands). Optimum cutting temperature compound was purchased from Sakura Finetek USA (Torrance, CA, USA). The Movat Pentachrom Kit was purchased from Morphisto GmbH (Offenbach am Main, Germany). ROTI Histokitt II was purchased from Carl Roth GmbH + Co. KG (Karlsruhe, Germany).

### Human aortic valve tissue samples
FCAVD tissues were obtained from 55 patients undergoing AV replacement for severe AS (Supplementary Data 1). Sex in the study participants was assigned at birth and gender identity was not considered. Tissue collection was approved by the local Ethical Committee (Medical Faculty, University Leipzig, registration number 128/19-ek) and all patients gave written informed consent in accordance with the Declaration of Helsinki. Using a stereoscopic microscope, tissue samples were macroscopically inspected and dissected into mildly diseased, fibrotic and calcific sections on-site and flash frozen in liquid nitrogen (Fig. 2a, Supplementary Fig. 7). The tissue allocation was confirmed histologically for an initial protocol establishment phase using Movat pentachrome staining as well as molecular imaging with a bisphosphonate based calcium tracer (OsteoSense) to visualize and confirm fibrosis and calcification. Deep-frozen tissue sections were grinded in a percussion mortar in liquid nitrogen and stored at −80 °C until further analysis.

### Pentachrome staining
Native AV samples were embedded perpendicularly to obtain longitudinal sections in OCT Embedding Matrix for Frozen Sections (Cell Path, Carl Roth #6478.1). Consecutive sections of 5 µm were cut using Cryostat (Leica CM1860, Germany) and mounted onto microscope slides (Superfrost Plus Slides, Epredia, USA). Cryosections were dried and fixed in Methanol/Aceton 1:1 solution. Movat Pentachrom Kit (Morphisto, Germany) was used according to the manufacturer's instructions. Following this staining, nuclei appear in blue-black, cytoplasm in light red, elastic and muscle fibers in red, collagenous connective tissue in light yellow, mineralized bone in dark yellow, and mineralized cartilage tissue in blue-green. The sections were mounted with xylene-containing mounting medium (ROTI Histokit II from Carl Roth #T160.1).

To quantify the calcification status of the samples, sections were incubated in phosphate-buffered saline containing 100 pmol/ml Osteosense solution (IVI SenseOsteo 680 #NEV10020Ex, PerkinElmer) for 24 h at 37 °C in the dark. Then, Hoechst 33342 (Miltenyi, Bergisch-Gladbach, Germany) was added in a 1000-fold dilution and incubated for another 10 min. Finally, sections were washed with water and the slides were mounted using Roti Mount FluorCare (Carl Roth #HP19.1). Sections were visualized using Keyence BZ-X800E (Keyence, Germany).

### Lipid extraction of pooled AV sample for lipid identification and ITSD design
All solvents used for the extraction were supplemented with 0.1% (w/v) BHT and cooled on ice before use. All extraction steps were performed on ice or at 4 °C. Pooled sample was formed by mixing equal amounts of AV tissue powders from each specimen (mildly diseased, fibrotic, and calcific). Lipids were extracted according to the Folch protocol[74]. 20 mg of pooled sample was placed into 2 mL tubes (Eppendorf, Hamburg, Germany) and 600 µL chloroform/methanol (2:1, v/v) was added. Samples were incubated for 1 h at 4 °C on a rotary shaker (300 rpm). 150 µL water was added to induce phase separation and the mixture was incubated for another 15 min at 4 °C on a rotary shaker (300 rpm). Samples were centrifuged ($10,000 \times g$, 10 min, 4 °C), the lower organic phase was transferred to a new tube and dried under vacuum.

### Untargeted lipidomics workflow for human AV lipids identification
Lipid extract obtained from the pooled AV sample was reconstituted in 200 µL isopropanol (Supplementary Fig. 8a). 5 µL (for analysis in positive ionization mode) and 10 µL (for analysis in negative ionization mode) were loaded on an Accucore reversed phase C30 column (2.1 × 150 mm, 2.6 µm, 150 Å; Thermo Fisher Scientific) installed on a Vanquish Horizon UHPLC (Thermo Fisher Scientific). UHPLC separation was coupled online either to Q Exactive Plus Hybrid Quadrupole Orbitrap (ID Method 1, 34 min gradient) or Exploris 240 Hybrid Quadrupole Orbitrap (ID Method 2, 57 min gradient) mass spectrometers, both equipped with a HESI source (Thermo Fisher Scientific).

ID Method 1, 34 min gradient: Lipids were separated by gradient elution with solvent A (acetonitrile/water, 1:1, v/v) and B (isopropanol/acetonitrile/water, 85:10:5, v/v/v), both containing 5 mM ammonium formate and 0.1% (v/v) formic acid. Separation was performed at 50 °C with a flow rate of 0.3 mL/min using the following gradient: 0−20 min−10−86% B, 20−22 min−86−95% B, 22−26 min−95% B, 26−26.1 min−95−10% B, followed by 7.9 min re-equilibration at 10% B. Mass spectra were acquired in positive and negative ionization modes using spray voltages of 3.5 kV and −2.5 kV, respectively, ion transfer temperature of 300 °C, aux gas heater temperature of 370 °C, S-lens RF level of 35%, auxiliary gas of 10 a.u., sheath gas of 40 a.u., and sweep gas of 1 a.u. In positive ion mode, full scan mass spectra were recorded from $m/z$ 250 to 1200 at a resolution of 140,000 at $m/z$ 200, AGC target of $1 \times 10^6$, maximum injection time of 100 ms, and default charge state of 1. In negative ion mode, scan ranges from $m/z$ 380 to 1200 or from $m/z$ 400 to 1600 were applied. Tandem mass spectra were recorded by data-

dependent acquisition (DDA) for the 15 most intense precursor ions (DDA top 15) at a resolution of 17,500 at $m/z$ 200, AGC target of $1 \times 10^5$, maximum injection time of 60 ms (negative mode: 150 ms), isolation window of 1.2 $m/z$ units, and stepped normalized collision energy (NCE; 10, 20, and 30). The DDA settings were set to an apex trigger of 6 s, isotope exclusion on, and dynamic exclusion for 10 s[24].

ID Method 2, 57 min gradient: Lipids were separated by gradient elution with solvent A (acetonitrile/water, 1:1, v/v) and B (isopropanol/acetonitrile/water, 85:10:5, v/v/v), both containing 5 mM ammonium formate and 0.1% (v/v) formic acid. Separation was performed at 50 °C with a flow rate of 0.3 mL/min using following gradient: 0–20 min– 10–80% B, 20–37 min–80–95% B, 37–41 min–95–100% B, 41–49 min– 100% B, 49–49.1 min–100%–10% B, followed by 7.9 min re-equilibration at 10% B. Mass spectra were acquired in positive and negative ionization modes using spray voltages of 3.5 kV and −2.5 kV, respectively, ion transfer temperature of 300 °C, aux gas heater temperature of 370 ∘C, S-lens RF level of 35%, auxiliary gas of 10 a.u., sheath gas of 40 a.u., and sweep gas of 1 a.u. In positive ion mode, full scan mass spectra were recorded from $m/z$ 200 to 1200 at a resolution of 120,000 at $m/z$ 200, AGC target of $1 \times 10^6$, maximum injection time was set to auto. In the negative ion mode, scan ranges from $m/z$ 200 to 1400 and $m/z$ 370 to 1480 with a maximum injection time of 100 ms were acquired. Easy-IC was set to RunStart for all methods. Tandem mass spectra were recorded by DDA within a cycle time of 1.3 s at a resolution of 30,000 at $m/z$ 200, an AGC target of $1 \times 10^5$, a maximum injection time of 54 ms, an isolation window of 1.2 $m/z$ units, a stepped NCE (17, 27, 37), and charge state 1. The DDA settings were set to isotope exclusion on, dynamic exclusion for 6 s (mass tolerance ±2.5 ppm, exclusion after 2 times within 10 s).

Software-assisted lipid identification was performed using Lipid Hunter2[75] (source code version: https://github.com/SysMedOs/lipidhunter). RAW files were converted with MSconvert (ProteoWizard version 3.0.9134) into mzML format. Lipids were identified using a mass accuracy of 5 and 20 ppm for MS1 and MS/MS spectra, respectively. To note, although 20 ppm mass accuracy was used for the annotation of MS/MS spectra, true experimental values were ≤5 and 10 ppm for analysis performed on Exploris 240 and Q Exactive Plus instruments, respectively. The remaining parameters were kept as default. Proposed lipid species were confirmed by manual inspection of the tandem mass spectra for species-specific fragment ions within the HTML report. CE, CL, and GM3 gangliosides were identified manually. Proposed identifications were further validated by plotting the retention time of lipid species against their Kendrick mass defect by hydrogen and lipid species not following expected trendlines were excluded. Details on lipid annotation are provided in the form of Lipidomics Minimal Reporting Checklists following the recommendations of the Lipidomics Standards Initiative (Supplementary note 1).

### Design of the AV-tailored mixture of internal standards (ISTD) for semi-absolute quantification

To design an ISTD mixture tailored to the lipid subclasses and their endogenous concentrations in human AV lipidome, the rough composition and relative abundance of AV lipids were first accessed via quantitative high-performance thin layer chromatography. To this end, pooled AV lipid extract (prepared as described above) was dissolved in 200 μL chloroform/methanol (2:1, v/v) and loaded onto a TLC plate (10 or 20 μL, HPTLC silica gel 60, 20 × 10 cm, Merck) using a Camag Linomat 5 sampler (Camag, Switzerland). On each plate, six-point serial dilutions of polar or apolar lipid TLC standards were loaded for quantitative lipid class-specific calibration (Supplementary Data 6). Plates were developed using chloroform/ethanol/triethyla-mine/water (5:5:5:1, v/v/v/v) or hexane/diethylether/acetic acid (85:15:1, v/v/v/v) for polar and apolar lipid quantification, respectively. Dried TLC plates were immersed in acetone/water (8:2, v/v) containing primuline (0.05%, w/v) for 5 s (Camag Chromatogramm Immersion

Device III). Analytes were visualized under UV light (366 nm) and scanned by a videodensitometric device (Biostep GmbH, Germany). Densitometric analysis was performed with Image Lab (Version 6.1, Bio-Rad). Using TLC-based quantification (Supplementary Fig. 9a, b), the rough abundances of major AV lipid classes were approximated, based on which the initial composition of ISTD mixture was designed by combining SPLASH® Lipidomix®: Cer/SpH Mix I in the ratio 1:1,5 (v/v), and additional CL and individual Cer standards were added (Supplementary Data 7).

For defining the appropriate amounts of ISTD mixture to be spiked into individual samples for quantification via single-point calibration, seven-point calibration curves were generated (Supplementary Fig. 8b). To this end, pooled AV tissue powder was spiked (each calibration point in independent triplicates) with selected ISTD mixture in seven different amounts (Supplementary Data 7). 21 samples were extracted as described above, extracts were reconstituted in 100 μL isopropanol, and analysed by LC-MS using C30 RPLC separation coupled online to Q Exactive Plus Hybrid Quadrupole Orbitrap mass spectrometer as described above with slight adaptations. Individual samples were recorded in full scan (MS1) positive and negative ion modes in the range from $m/z$ 100 to 1500. Additionally, 10 μL of each sample was combined into a total quality control (tQC) sample, which was acquired in DDA as described above in order to map lipid annotations to full scan mode (MS1) data when necessary.

Peak areas of five to nine endogenous lipids per subclass (selected from the identification Supplementary Data 2) representing the highest, the middle, and the least abundant species, as well as the corresponding ISTD, were integrated using Skyline (version 22.2) software[76]. Area under the curve (AUC) for different class-specific adducts and common in-source fragments (see below for more details) was summed up. Incomplete isotopic enrichment was corrected for deuterated ISTD. All peaks were subjected to type I isotopic corrections[77].

For each ISTD, seven-point calibration curves were generated (Supplementary Fig. 10), and the linearity range of LC-MS responses was accessed. Additionally, peak areas for each ISTD were plotted relative to the peak areas of the endogenous AV lipids to select the ISTD amounts not only within the linear range of LC-MS responses but also preferentially representing the middle range of the endogenous lipid concentrations for each considered lipid subclass (Supplementary Fig. 11).

The final AV-tailored ISTD mixture was prepared by combining individual standards from the stock solutions in the required amounts (Supplementary Data 8). Complete ISTD mixture was prepared once in the quantities sufficient for all individual samples, dried under vacuum, re-dissolved in chloroform/methanol (2:1, v/v) and divided into working aliquots (one per extraction batch) that were dried, stored at −80 °C, and reconstituted in chloroform/methanol (2:1, v/v) before use. To each sample corresponding roughly to 20 mg of AV tissue, 10 μL of ISTD mixture with individual standards amount provided in Supplementary Data 8 was added before the lipid extraction.

### Extraction and untargeted lipidomics workflow for human AV lipids quantification

All individual samples were extracted and analysed in a randomized order to account for possible batch effects during sample preparation and LC-MS experiments (Supplementary Fig 8d). For the evaluation of potential batch effects, representative pools were created by combining sample material from the same sample cohort (batch quality control, BQC). Samples were randomly grouped into batches of a maximum of 35 samples per batch. The BQC samples were split into aliquots of 20 mg each and distributed to the different batches. Blank (without biological material, used to account for a chemical noise generated from extraction solvents and materials), individual samples, and BQCs of the same batch were extracted simultaneously. All solvents used for extraction were supplemented with 0.1% (w/v) BHT and

cooled on ice before use. All extraction steps were performed on ice or at 4 °C. Samples were allowed to thaw on ice for 30 min prior to extraction. To ~20 mg of each individual AV sample tissue powders 600 μL chloroform/methanol (2:1, v/v) and 10 μL of ISTD mixture in chloroform/methanol (2:1, v/v; Supplementary Data 6) were added, mixed, and incubated on ice for 15 min. Samples were incubated for 1 h at 4 °C on a rotary shaker (300 rpm). 150 μL water was added to induce phase separation and the mixture was incubated for 15 min at 4 °C on a rotary shaker (300 rpm). Samples were centrifuged (10,000 × g, 10 min, 4 °C), the lower organic phase was transferred to a new tube, and dried under vacuum.

Lipid extracts were reconstituted in 110 μL isopropanol and 2 μL were loaded on an Accucore reversed phase C30 column (2.1 × 150 mm, 2.6 μm, 150 Å; Thermo Fisher Scientific) installed on a Vanquish Horizon UHPLC (Thermo Fisher Scientific) coupled online to an Exploris 240 Hybrid Quadrupole Orbitrap mass spectrometer. Acquisition sequence was designed as described previously[78], and included column equilibration, acquisition of blankQC (blank extracts), column conditioning with tQC, tQC dilutions to control for the possible carryover and the linearity of the MS response, BQCs, and randomized individual samples and tQC samples injected every 10 runs to control for LC-MS performance. Lipids were separated by gradient elution with solvent A (acetonitrile/water, 1:1, v/v) and B (isopropanol/acetonitrile/water, 85:10:5, v/v/v), both containing 5 mM ammonium formate and 0.1% (v/v) formic acid. Separation was performed at 50 °C with a flow rate of 0.3 mL/min using following gradient: 0–10 min–30–80% B, 10–27 min–80–95% B, 27–31 min–95–100% B, 31–39 min–100% B, 39–39.1 100–10% B, followed by 7.9 min re-equilibration at 10% B. Mass spectra were acquired in positive mode using spray voltages of 3.5 kV, ion transfer temperature of 300 °C, aux gas heater temperature of 370 °C, S-lens RF level of 35%, auxiliary gas of 10 a.u., sheath gas of 40 a.u., and sweep gas of 1 a.u. Easy-IC was set to RunStart. Mass spectra were recorded in full scan (MS1) from $m/z$ 200 to 1200 at the resolution of 120,000 at $m/z$ 200, AGC target of $1 \times 10^6$, maximum injection time auto. Additionally, 10 μL of each sample was combined into a tQC sample, which was acquired after every 10 samples as well as at the start of the sequence in DDA mode (positive and negative ionization modes; described above) in order to map lipid annotations to full scan mode (MS1) data if necessary.

Lipids were quantified using Skyline (version 22.2)[76]. AUC for different class-specific adducts and common in-source fragments (Supplementary Data 9) were summed up. Incomplete isotopic enrichment was corrected for deuterated ISTD. All signals were subjected to type I isotopic corrections[77]. Individual lipid species were quantified in relation to the respective lipid class-specific ISTD and normalized to the wet tissue weight. Linear regression analysis was applied by plotting the calculated concentrations against their AUC value to identify and exclude possible outliers and features showing nonlinear behaviour. Details on lipid quantification are provided in the form of Lipidomics Minimal Reporting Checklists following the recommendations of the Lipidomics Standards Initiative (Supplementary note 1).

### Assessment of PA and LPA recovery rates

To determine the recovery of PA and LPA lipids, pooled AV tissue powders were spiked with selected PA/LPA ISTD (LPA 18:1, PA 17:0/17:0, PA 18:0/18:0, PA 18:0/18:2) with the amounts corresponding to 0, 10, 100, and 1000 pmol per 20 mg of tissue (in triplicates) before and post lipid extraction (Supplementary Fig. 8c). Lipid extraction was performed using a chloroform/methanol/water mixture (Folch extraction; as described above) or by acidified methanol/chloroform extraction[36]. Briefly, for acidified methanol/chloroform extraction, 20 mg of pooled AV were mixed with 800 μL ice-cold methanol/0.1 M HCl in water (1:1; v/v). 10 μL of PA/LPA ISTD in chloroform/methanol (2:1, v/v) were added, mixed and incubated for 15 min on ice. Next, phase separation was induced by adding 400 μL of chloroform, samples were centrifuged (10,000 × g, 10 min, 4 °C), lower organic phase

was transferred to a new tube and dried under vacuum. All solvents used for the extraction were supplemented with 0.1% (w/v) BHT and cooled on ice before use. All extraction steps were performed on ice.

Lipid extracts were reconstituted in 100 μL methanol/isopropanol (1/1, v/v), 5 μL were loaded on an Accucore reversed phase C30 column (2.1 × 150 mm, 2.6 μm, 150 Å; Thermo Fisher Scientific) installed on a Vanquish Horizon UHPLC (Thermo Fisher Scientific). Lipids were separated by gradient elution with solvent A (acetonitrile/water, 1:1, v/v) and B (isopropanol/acetonitrile/water, 85:10:5, v/v/v) both containing 5 mM ammonium formate and 0.1% (v/v) formic acid. Separation was performed at 50 °C with a flow rate of 0.3 mL/min using following gradient: 0–20 min–10–86% B, 20–22 min–86–95% B, 22–26 min–95% B, 26–26.1 min–95–10% B, followed by 7.9 min re-equilibration at 10% B. Mass spectra were acquired on Exploris 240 operating in negative ionization modes using spray voltage of −2.5 kV, ion transfer temperature of 300 °C, aux gas heater temperature of 370 °C, S-lens RF level of 35%, auxiliary gas of 10 a.u., sheath gas of 40 a.u., and sweep gas of 1 a.u. Easy-IC was set to RunStart. Full scan mass spectra were recorded from $m/z$ 200 to 1200 at a resolution of 120,000 at $m/z$ 200, AGC target of $1 \times 10^6$, maximum injection time of 100 ms, and default charge state of 1. Tandem mass spectra were recorded by DDA within a cycle time of 1.3 s at a resolution of 15,000 at $m/z$ 200, an AGC target of $1 \times 10^5$, a maximum injection time of 60 ms, an isolation window of 1.2 $m/z$ units, and a stepped NCE (17, 27, 37), isotope exclusion on, and dynamic exclusion for 6 s. Lipids were quantified using Skyline (version 22.2) as described above. Recovery rates were determined by dividing the mean AUC of the samples spiked with PA/LPA ISTD before and post extraction, multiplied by 100%. Details on lipid annotation and quantification are provided in the form of Lipidomics Minimal Reporting Checklists following the recommendations of the Lipidomics Standards Initiative (Supplementary notes 2, 3).

### Extraction and targeted lipidomics workflow for human AV lipids quantification

For quantification of LPA, LPI, LPS, PA, PI, PS, PG, CL, and GM3 lipids, acidified extraction and targeted LC-MS/MS analysis (selected reaction monitoring, SRM; see details below) in negative ionization mode was used. First, to determine the appropriate amounts of ISTD, a seven-point calibration curves were generated using the workflow described above (Supplementary Data 5). Linearity of the response was evaluated and the final amount of ISTD to be spiked in 20 mg of AV tissue powder was selected (Supplementary Data 6). ~20 mg of each individual sample (Supplementary Data 1) was spiked with selected ISTD amounts and used for the acidified extraction as described above (Supplementary Fig. 8e). Lipid extracts were reconstituted in 100 μL methanol/isopropanol (1:1, v/v) and 3 μL were loaded on an Accucore reversed phase C18 column (2.1 × 150 mm, 2.6 μm, 80 Å; Thermo Fisher Scientific) installed on a Vanquish Flex UHPLC (Thermo Fisher Scientific) coupled online to TSQ Altis Plus Triple Quadrupole mass spectrometer equipped with a HESI source (Thermo Fisher Scientific). Lipids were separated by gradient elution with solvent A (acetonitrile/water, 1:1, v/v) and B (isopropanol/acetonitrile/water, 85:10:5, v/v/v), both containing 5 mM ammonium formate and 0.1% (v/v) formic acid. Separation was performed at 50 °C with a flow rate of 0.3 mL/min using the following gradient: 0–7 min–30–85% B, 7–8 min–85 to 95% B, 8–10 min–95% B, 10–10. 1 min–95–30% B, followed by 4.9 min re-equilibration at 10% B. Mass spectra were acquired in selected reaction monitoring (SRM) mode using spray voltages of −2.5 kV, ion transfer temperature of 300 °C, vaporizer temperature of 370 °C, auxiliary gas of 10 a.u., sheath gas of 50 a.u., and sweep gas of 1 a.u. SRMs were recorded with setting automatic dwell time to ensure acquisition of 10 points per peak, dwell time factor 3, Q1 and Q3 resolution (FWHM) of 0.7 Da, and CID gas flow of 1.5 mTorr. Lipid-species-specific transitions, collision energy, and RF lens voltage for each analyte are listed in the Supplementary Data 10. For quantification, data were processed using

Skyline (version 22.2) as described above. Both type I and II isotopic corrections were applied and individual lipid species were quantified by the respective lipid class-specific ISTD and normalized to the sample weight. Linear regression analysis was applied by plotting the calculated concentration of lipid species against their AUC value to identify and exclude outliers and features with nonlinear behaviour.

## Statistical analysis

Quantitative data analysis, including isotopic corrections, ISTD normalization, and normalization by sample weight, was performed with Microsoft Excel 2016. Graphical representations were generated using Corel DRAW 2020 22.0 (Corel corporation), Graphpad Prism® 8.0.2 (GraphPad Software, Inc.), Inkscape 1.3.2 (Inkscape Developers), Python 3.12 (Python Software Foundation), and R Statistical Software v4.4.1(R Core Team 2021). Statistical analysis was performed using Graphpad Prism® 8.0.2, MetaboAnalyst version 6.0[79], Python 3.12, and R Statistical Software v4.4.1. Statistical significance between compared groups was found by $t$ test or with an ANOVA with a threshold of $P \leq 0.05$. Generic Python libraries for data processing and visualization tasks, such as Pandas, Scikit-learn, Matplotlib, and Plotly, are used in different plots. The detailed configurations and versions can be found in the corresponding source code repositories.

## Lipid distribution plot

The lipid quantification results were averaged and used for the generation of lipid distribution plot using customized Python scripts. Inspired by the distribution plot published by Burla et al.[80], a python version of the distribution plot was developed and used for the successful visualization of adipocyte's lipidome for the AdipoAtlas project[24] (https://github.com/SysMedOs/AdipoAtlasScripts). The code takes the concentration of each lipid, plots a colour bar on the log scale according to the corresponding lipid subclass. A bold colour bar representing the total sum of each lipid subclass is plotted automatically. Based on the previous code base, the improved version to visualize the distribution of lipids is summarized into the LipidomeDistribution repository, and the source code is released under AGPL v3 license on GitHub: https://github.com/LMAI-TUD/LipidomeDistribution.

## Uniform manifold approximation and projection (UMAP) algorithm

To generate UMAP, lipid quantities, fatty acyl chains composition (carbon and double bond count) and myopic MCES distances for lipid class-specific structures were considered. Briefly, SMILES representation of the lipid class-specific backbone (lipid_class_backbone_smiles) without FA residues was generated, and the myopic MCES distances to other lipid_class_backbone_smiles were calculated using the myopicmces package[81] (source code: https://github.com/boecker-lab/myopic-mces). The resulting myopic MCES distances matrix (24 × 24) was used to perform a PCA (number of components = 10) to reduce data dimensionality, and the top 5 components were further used for UMAP analysis. To bring all used values in the similar range, lipid concentrations were z-scored, and all other values were scaled between 0 and 1 using a min-max scaler by the Scikit-learn Python library. Different weight factors were applied to the scaled values to generate a final data matrix for the UMAP representation. A detailed description of all parameters was summarized in the Supplementary Data 11. The source code of LipidomeUMAP is released under the AGPL v3 license on GitHub: https://github.com/LMAI-TUD/LipidomeUMAP.

## Sankey plot

Sankey plot for the visualization of Sphingolipids was previously introduced by the AdipoAtlas project[24] (https://github.com/SysMedOs/AdipoAtlasScripts). In brief, for each sphingolipid, the sphingoid base, the FA residue, and the corresponding sphingolipid subclass are assigned and connected in the concept of the Sankey flow diagram. The widely used visualization library Plotly for Python was used to generate the Sankey plot from the identified sphingolipids. The updated Sankey plot for this project is released under AGPL v3 license on GitHub: https://github.com/LMAI-TUD/LipidSankey.

## Lipid trends analysis

Trend analysis diagrams were generated by plotting the individual lipid clusters from Gaussian Mixture Model clustering of lipids according to their concentration variances across the mildly diseased, fibrotic, and calcific stages. The missing values (39 out of 24,225 values; 0.16% of total values) in the quantified data matrix table were filled with 1/5 of the minimum value of the corresponding lipid across all samples according to the widely used MetaboAnalyst tool[79]. The filled data matrix was averaged to the mildly diseased, fibrotic, and calcific stages. Z-score scaling followed by Gaussian Mixture Model clustering was performed on the averaged data matrix using the mainstream scikit-learn Python library. The clustered results were then plotted as a trend plot for each cluster by using matplotlib Python library. After evaluation of different combinations of scikit-learn built-in scaling methods (min-max, Z-score, and Log₂) with mainstream clustering algorithms (Hierarchical clustering, K-means, bisecting K-means, Gaussian Mixture Model, and Dirichlet Process Gaussian Mixture Model), the Z-score and Gaussian Mixture Model combination was selected for the generation of the plots. The reusable Python scripts providing generic access to multiple scaling and clustering algorithms are provided on GitHub under AGPL v3 license: https://github.com/LMAI-TUD/LipidTrends.

## Heatmaps

Heatmaps were generated using the gplots R package[82] by plotting the individual lipid concentrations of significantly regulated lipids (determined by ANOVA $P < 0.05$) for male and female patients across the mildly diseased, fibrotic, and calcific stages for individual samples or group means using customized R scripts. The missing values (39 out of 24,225 values; 0.16% of total values) in the quantified data matrix table were filled with 1/5 of the minimum value of the corresponding lipid across all samples according to the widely used MetaboAnalyst tool[79]. Lipid concentration values were log-transformed, mean-centered, and divided by the standard deviation of each variable.

## Reporting summary

Further information on research design is available in the Nature Portfolio Reporting Summary linked to this article.

## Data availability

The data used for above mentioned plots can be accessed in the corresponding GitHub code repositories from the LMAI team https://github.com/LMAI-TUD. The data supporting the findings from this study are available within the manuscript and its supplementary information. The raw data/mzML data generated in this study have been deposited in the Metabolomics Workbench under accession code ST003364 (doi: 10.21228/M8G24F)[23]. Source data are provided with this paper.

## Code availability

The source code used for the above-mentioned plots can be accessed in the corresponding GitHub code repositories from the LMAI team https://github.com/LMAI-TUD.

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

## Acknowledgements

This study was supported by the German Research Council (DFG Schi 476/19-1 and Schi 476/19-1 as well as SFB 1052/Z3 to J.S.). Work in the Fedorova lab is supported by "Sonderzuweisung zur Unterstützung profilbestimmender Struktureinheiten" by the SMWK to TUD, TG70 by Sächsische Aufbaubank and SMWK, the measure is co-financed with tax

funds on the basis of the budget passed by the Saxon state parliament (to M.F.), Deutsche Forschungsgemeinschaft (FE 1236/5-1, FE 1236/8-1 to M.F.), and Bundesministerium für Bildung und Forschung (031L0315A, DEEP_HCC and 01EJ2205A, FERROPath to M.F.).

## Author contributions

Conceptualization and study design were carried out by P.P., J.S., F.S., and M.F. Initial collection of sample material and preparation was performed by J.B., S.W., H.T., P.B., and F.S. Sample preparation for mass spectrometry experiments and data acquisition was performed by P.P. and M.W. J.B., S.W., and P.B. performed histological staining. Analysis and interpretation of data were carried out by P.P., M.W., Z.N., and M.F. P.P. and M.F. wrote the manuscript with contributions from all co-authors.

## Funding

## Competing interests

The authors declare no competing interests.
