## [Transparent Peer Review file · Nature Communications]

Sex-specific lipidomic signatures in aortic valve disease reflect differential fibro-calcific progression

Corresponding Author: Professor Maria Fedorova

Version 0:

Reviewer comments:

Reviewer #1

(Remarks to the Author)

I'd like to congratulate the authors to a well structured study and their lipidomics research. I find some of the terms and strategies not fitting for such kind of work and would like to see this addressed in a revised version.

However, some critical points should be addressed in my opinion:

1. I don't know what a "semi-quantitative molecular description of the human AV lipidome" would be worth for the community. And I also find "lipids were semi-absolutely quantified" misleading. These are not well defined terms. You have chosen a quantitation strategy, which is ok described. Please define in the discussion section the limits of this approach.

1a) Please tone down the claim of a "reference lipidome" when it's not quantified with the according confidence.

1b) I have problems to recognize what method is now used for the lipidome analyses. Different gradients were used and one has to be clear here and consistent.

1c) Please describe in a supplement document how you have ensured that no memory effects on the column occur and how you have controlled for lipid background originating from used consumables. Were blank extraction analyzed as control samples?

1d) The way how pools were generated is not clear to me. How many biological replicates (pools) are in the end compared for this rather small cohort, which is still fine for this kind of specimen?

1e) I find mass accuracy settings of 20ppm in MS2 for utilized Orbitrap type instruments far too relaxed and can not see a reason for that choice.

1f) I would appreciate the use of the Lipidomics Minimal Reporting list for this study.

2. PCA - plots (Figure 3, 4) are not the right tools to describe global trends or shifts in a lipidome. I would recommend an analysis based on proper statistics using ANOVA and enrichment analyses that also provide better insights in the underlying metabolic pathways. PCA - Plots have to include the biological replicates and also indication of variability within a group using confidence areas.

a) Filling missing values with "1/5 of the minimum value of the corresponding lipid across all samples ..." is in my assessment not the right way and it has a bias. Please state from which occupation threshold (% samples with experimental value) you have filled in missing values? Please use a random forest model for comparison.

3. I find many of the figures lack annotations that help the reader recognizing certain claims in the text. Specifically, I would appreciate usage of fatty acyl/alkyl level annotation where possible. Figure 3g-i / Figure 4 a-c,e. For lipids with significant changes some representative should be chosen and shown.

Reviewer #2

(Remarks to the Author)

The authors performed quantitative lipidomics to profile the metabolic trajectories in human tricuspid and bicuspid aortic valves, and found stage dependent extrinsic and intrinsic lipid trends. Major extrinsic lipids derived from infiltrating lipoproteins and were further metabolized within the AV. Intrinsic lipid remodeling suggested tissue degeneration with a loss of phosphatidylserines. Male and female patients showed markedly different lipid signatures of FCAVD progression. The high extent of sexual dimorphism in the valve lipidome strongly suggests that tailored approaches for pharmacological

intervention. This is an interesting study with a considerable amount of work on an important and timely topic. There are however several issues to consider:

1- The authors report substantial differences between BAV vs. TAV and also between men vs. women. However, as shown in Supplementary Table 1, these results are derived from small subgroups ($n < 15$ per group and < 10) and with important differences in age and most likely in comorbidities and cardiovascular risk factors (not shown) between groups. Age is a major confounding factor in these analyses. Age likely has a major effect on lipidomics and valve lipidome in particular. There is about 10 years difference between BAV and TAV. So it is impossible to determine to which extent the differences in valve lipidome between BAV vs. TAV is related or not to age.

2- It is likely that men and women subgroups were not well balanced in terms of age and other cardiometabolic risk factors, which may have also affected the results on sex related differences. The authors would need to add a comprehensive table showing and comparing the age and other baselines characteristics of the patients in the different subgroups. This table should include, age, sex, BMI, dyslipidemia, hypertension, diabetes, coronary artery disease, pharmacotherapy at the time of valve explant (statins, anti-hypertensive therapy etc.).

3- Pharmacotherapy that was present at the time of valve explant may have affected the results of valve lipidome, and in particular therapies targeting lipids and diabetes. This information was not provided in the paper and it is likely that the different BAV vs. TAV, men vs. women and stage of disease were not well matched for age, cardiometabolic risk factors, and pharmacotherapies.

4- The methods on the lipidomic analyses is very well described and detailed. On the other hand, the section on Human aortic valve tissue samples is described briefly in a short paragraph. Still this section is absolutely essential. The matching of the comparison groups for age, cardiometabolic risk factors, and pharmacotherapies is key for the validity and the interpretability of the inter-group comparisons.

5- The methods used to classify determine the status of the lesion: mildly diseased, fibrotic and calcific is not described.

6- The clinical implications of the results of this study are unclear and would merit to be better described.

Version 1:

Reviewer comments:

Reviewer #1

(Remarks to the Author)

The authors have addressed my points comprehensively regarding documentation and sharing of the lipidome data. Biological and clinical interpretation remain difficult for me to judge. In the current form the manuscript provides important research data for the community and should be checked and evaluated by follow up studies. I appreciate the clear and scientifically sound responses to critical methodical questions. From my perspective, the manuscript can be accepted for publication.

Reviewer #2

(Remarks to the Author)

The authors have well addressed my previous comments and the revised version is further improved. It would be useful to include a central illustration or visual abstract that would synthesize and summarize the key messages of this study.

Point-by-point answers to the Reviewer(s)' Comments:

Reviewer: 1

I'd like to congratulate the authors to a well-structured study and their lipidomics research. I find some of the terms and strategies not fitting for such kind of work and would like to see this addressed in a revised version.

We are grateful to the Reviewer for the positive feedback on our manuscript, as well as for the constructive criticism, which helped to significantly improve the quality of the results presentation.

However, some critical points should be addressed in my opinion:

1. I don't know what a "semi-quantitative molecular description of the human AV lipidome" would be worth for the community. And I also find "lipids were semi-absolutely quantified" misleading. These are not well defined terms. You have chosen a quantitation strategy, which is ok described. Please define in the discussion section the limits of this approach.

Following this valid suggestion, we removed terms "semi-quantitative" from the revised version of the manuscript. We specified how quantification was performed in the Results section of the manuscript and refer to the detailed description of performed experiments in Methods section where we provide information on the rational design of the AV-specific mixture of internal standards (both qualitatively in terms of the selected lipid subclasses as well as quantitatively in terms of the amounts used for the spiking), calibration strategy, ionization adducts used for the quantification as well as performed isotopic corrections.

"Reverse phase chromatography (RPC) coupled online with mass spectrometry (MS) analysis performed in an untargeted and targeted manner (see Methods section for details) allowed to identify 1073 lipid molecular species (Supplementary Table 2), of which 480 lipids were **semi-absolutely** quantified **using class-matching internal standards** (Supplementary Table 3 and Fig. 1a)."

1a) Please tone down the claim of a "reference lipidome" when its not quantified with the according confidence.

We removed the term "reference lipidome" from the revised version of the manuscript.

"Finally, all raw and processed lipidomics data produced in this study are made publicly available via Metabolomics Workbench²² resource, representing the first comprehensive

human AV reference lipidome available for the scientific community for further analysis and data interrogation²³. “

1b) I have problems to recognize what method is now used for the lipidome analyses. Different gradients were used and one has to be clear here and consistent.

Thank you for your valuable comment. We agree that the complexity of the experimental approach, along with the varied analytical strategies used for identification and quantification, made it challenging to follow up on some aspects. In the revised version of the manuscript, we introduced a Supplementary Figure 8, which provides a schematic overview of the experimental design and analytical approach. Different panels (a-e) of the Supplementary Figure 8 are now cross-referenced to the corresponding parts of the Methods section, which, we hope, will improve the clarity and the readability of the manuscript.

Supplementary Fig. 8 | Schematic overview of the experimental design used for the lipidomics analysis. Sample types, extraction methods, LC and MS acquisition types listed for identification (a), calibration (b and c), and quantification (d and e) experiments. DDA – data-dependent acquisition; ISDT – internal standards; LPA – lysophosphatidic acid, PA - phosphatidic acid; SRM – single reaction monitoring.

	Identification		Calibration		Quantification	
	a) Untargeted lipidomics for human AV lipids annotation		b) Design of the AV-tailored mixture of ISTD for quantification	c) Assessment of PA and LPA recovery	d) Untargeted lipidomics for human AV lipids quantification	e) Targeted lipidomics for human AV lipids quantification
Sample type	pooled AV sample		pooled AV sample and ISTD		I. individual samples with added one-point ISTD mixture II. batch quality control (BQC) (pooled from AV samples) III. total quality control (tQC) pooled from individual AV samples	I. individual samples with added one-point ISTD mixture II. batch quality control (BQC) (pooled from AV samples) III. total quality control (tQC) pooled from individual AV samples
Extraction	Folch	Folch	Folch	Folch, acidified MeOH/CHCl ₃	Folch	Folch, acidified MeOH/CHCl ₃
LC	RPC30 method 1 (34 min gradient)	RPC30 method 2 (57 min gradient)	RPC30 method 1 (34 min gradient)	RPC30 method 1 (34 min gradient)	RPC30 method 3 (47 min gradient)	RPC18 method 4 (15 min gradient)
MS	DDA (Q Exactive Plus Hybrid Quadrupole Orbitrap)	DDA (Exploris 240 Hybrid Quadrupole Orbitrap)	MS1 (Q Exactive Plus Hybrid Quadrupole Orbitrap)	DDA (Exploris 240 Hybrid Quadrupole Orbitrap)	MS1 (Exploris 240 Hybrid Quadrupole Orbitrap)	SRM (Altis Plus Triple Quadrupole)

1c) Please describe in a supplement document how you have ensured that no memory effects on the column occur and how you have controlled for lipid background originating from used consumables. Were blank extraction analysed as control samples?

Indeed, this is a crucial analytical consideration for quantitative analyses. In the revised version of the manuscript, we provide additional information on the experimental design used in the study aiming to ensure no memory effects and linearity of the MS responses used for lipid quantification. We also cite an additional reference (PMID 39354313) to the recent article which provide a step-by-step detailed description of the procedures regularly implemented in our laboratory to design the LC-MS analysis aiming for the lipid quantification in large cohorts of samples, including all types of blank and Quality Control samples.

“Blank (without biological material, used to account for a chemical noise generated from extraction solvents and materials), individual samples and BQCs of the same batch were extracted simultaneously.”

„Acquisition sequence was designed as described previously⁷⁸, and included column equilibration, acquisition of blankQC (blank extracts), column conditioning with tQC, tQC dilutions to control for the possible carryover and the linearity of the MS response, BQCs, and randomized individual samples and tQC samples injected every 10 runs to control for LC-MS performance.”

1d) The way how pools were generated is not clear to me. How many biological replicates (pools) are in the end compared for this rather small cohort, which is still fine for this kind of specimen?

We hope that newly introduced Supplementary Figure 8 described in the comment above, will clarify the use of pooled samples. Briefly, as illustrated by Supplementary Figure 8, pooled AV samples were used for lipid identification (Figure S8a) and calibration (Figure S8b and c) purposes. Additionally, pooled samples were employed to generate batch quality control (BQC) and total quality control (tQC) samples used to control for the possible variability during sample preparation and LC-MS analysis, respectively. All quantitative results reporting differences in the lipidome of TAV and BAV samples along FCAVD stages, are derived from the analysis of individual, non-pooled samples.

1e) I find mass accuracy settings of 20ppm in MS2 for utilized Orbitrap type instruments far to relaxed and can not see a reason for that choice.

We examined the lipid identification reports and indeed can confirm that true, experimental mass accuracy was much below 20 ppm. Thus, to address this comment, we provided an additional statement in the corresponding part of the Methods section.

„Lipids were identified using a mass accuracy of 5 and 20 ppm for MS1 and MS/MS spectra, respectively. To note, although 20 ppm mass accuracy was used for the

annotation of MS/MS spectra, true experimental values were ≤ 5 and 10 ppm for analysis performed on Exploris 240 and Q Exactive Plus instruments, respectively.”

1f) I would appreciate the use of the Lipidomics Minimal Reporting list for this study

Following the Reviewer suggestions, we provide now three Lipidomics Minimal Reporting Checklists following the recommendations of Lipidomics Standards Initiative, referenced as Supplementary Files 1-3.

Supplementary File 1 | Lipidomics Minimal Reporting Checklist with details on the AV lipid annotation and quantification using untargeted lipidomics workflow.

Supplementary File 2 | Lipidomics Minimal Reporting Checklist with details on the AV lipid quantification using Folch extraction method and targeted lipidomics workflow.

Supplementary File 3 | Lipidomics Minimal Reporting Checklist with details on the AV lipid quantification using acidified methanol/chloroform extraction method and targeted lipidomics workflow.

1. PCA - plots (Figure 3, 4) are not the right tools to describe global trends or shifts in a lipidome. I would recommend an analysis based on proper statistics using ANOVA and enrichment analyses that also provide better insights in the underlying metabolic pathways. PCA - Plots have to include the biological replicates and also indication of variability within a group using confidence areas.

Following the Reviewer suggestions, we removed PCA plots (Figure 3 and 4) from the revised version of the manuscript. Instead, as recommended, we performed ANOVA analysis and visualized significantly regulated lipids using heatmaps. New figures and the corresponding results and discussion are now provided in the revised version of the manuscript. Heatmaps on Figure 3e and 4d report average values per group and new Supplementary Figures 5 and 6 report values for the individual samples. We additionally list all significantly regulated lipids based on the ANOVA analysis in the newly introduced Supplementary Table 4.

Supplementary Table 4 | One-way ANOVA & post-hoc results comparing TAV and BAV lipidomes along different stages of FCAVD development. a, statistical comparison of mildly diseases, fibrotic and calcified lipidomes of TAV; **b**, statistical comparison of male and female mildly diseases, fibrotic and calcified lipidomes of TAV; **c**, statistical comparison of male and female mildly diseases, fibrotic and calcified lipidomes of TAV and BAV.

Updated text with the Results section referring to the changes introduced in the Figure 3.

“To take a deeper look into sex-specific FCAVD lipid phenotypes, we tracked lipidome-remodeling at different pathological stages in a sex-specific manner (Fig. 3e and Supplementary Fig. 5). In total, 283 lipids showed significant differences (ANOVA $p < 0.05$; Supplementary Table 4b) between mildly diseased, fibrotic and calcified TAV of male versus female individuals. Although many lipid alterations represented the major FCAVD development trends discussed above, female patients showed much higher accumulation of lipoprotein-derived lipids (CE, PC, SM) as well as the products of their metabolic processing (LPC, Cer) along the FCAVD progression. The major sex-specific differences between the pathological stages seems to be driven mainly by CE and SP in females and TG lipids in males. Indeed, TG lipids, did not show any significant differences in FCAVD progression when male and female samples were analyzed together, and emerge as one of the main sex-specific trends already in the mildly diseased stage.”

Updated Figure 3.

Fig. 3 | Fibro-calcific remodeling of aortic valve lipidome is sex-specific. **a**, Extracellular matrix remodeling in female and male FCAVD TAV sections visualized with Movat's pentachrome staining (top panels) shows collagen (yellow) and glycosaminoglycans (turquoise) and OsteoSense and DAPI staining (bottom panels) for calcific mineralization (red) and nuclei (blue), respectively. Bar correspond to 100 μm . **b-d**, Bar plots showing the mean total concentration of cholesteryl esters (CE) (**b**), triacylglycerides (TG) and free cholesterol (ST) (**c**), glycerophospholipids (GPL) and sphingolipids (SP) (**d**) in mildly diseased, fibrotic and calcific TAV tissue sections of male and female patients, respectively. Margins indicate standard deviation. **e**, Heatmap illustrating normalized abundances of the significantly regulated lipids (ANOVA $p < 0.05$) in TAV lipidomes (T) of female (f) and male (m) individuals across different pathophysiological stages of FCAVD (md - mildly diseased; fib - fibrotic; cal - calcific). **f-h**, Volcano plots illustrating lipid species significantly regulated between female versus male individuals in mildly diseased (**f**), fibrotic (**g**) and calcific (**h**) TAV tissue sections determined by fold change (FC > 1.5) analysis and T-test ($P < 0.05$).

Corresponding Supplementary Figure 5.

Supplementary Fig. 5 | Heatmap illustrates TAV lipidomes of female and male individuals across different pathophysiological stages of FCAVD. Heatmap only includes lipid species significantly regulated between female versus male individuals across different pathophysiological stages, including mildly diseased, fibrotic and calcific in TAV tissue sections determined One-way ANOVA & post-hoc ($P < 0.05$).

Updated text with the Results section refereeing to the changes introduced in the Figure 4.

“Contrary to the previous observations in TAV, heatmap representation of significantly regulated lipids (361 lipids; ANOVA $p \leq 0.05$; Supplementary Table 4c) showed that BAV display less sex-driven dimorphism of disease progression. Accumulation of lipoprotein-derived lipids (CE, PC, SM) and products of their metabolic processing (LPC, Cer) showed quite similar trajectories in male and female BAVs (Fig. 4d and Supplementary Fig. 6). Whereas, female TAVs showed higher levels of SP, and male TAV had higher content of TG relative to BAVs at all diseases stages. It is important to note that individuals with BAVs typically experience an earlier onset of FCAVD, with clinically manifested symptomatic AS occurring on average 10 years earlier than in patients with the physiological TAV. Subsequently, the observed variability in lipidomes between TAV and BAV may be attributed to differences in the age of patients with bicuspid versus tricuspid aortic valves.”

Updated Figure 4.

Fig. 4 | Lipidomics signatures discriminate tricuspid (TAV) from bicuspid (BAV) AV in FCAVD. a-c, Volcano plots illustrating lipid species significantly regulated between TAV versus BAV lipidomes of mildly diseased (**a**), fibrotic (**b**) and calcific (**c**) AV tissue sections determined by fold change ($FC > 1.5$) analysis and T-test ($P < 0.05$). **d,** Heatmap illustrating normalized abundances of significance regulated lipids (ANOVA $p < 0.05$) of TAV (T) and BAV (B) lipidomes of female (f) and male (m) individuals across different pathophysiological stages of FCAVD (md - mildly diseased; fib - fibrotic; cal - calcific). **e,** FCAVD progression is characterized by the same lipidome alterations in TAV and BAV.

Corresponding Supplementary Figure 6.

Supplementary Fig. 6 | Heatmap illustrates TAV and BAV lipidomes of female and male individuals across different pathophysiological stages of FCAVD. Heatmap only includes lipid species significantly regulated between female versus male individuals across different pathophysiological stages, including mildly diseased, fibrotic and calcific in TAV and BAV tissue sections determined One-way ANOVA & post-hoc ($P < 0.05$).

We also tried to perform the enrichment analysis suggested by the Reviewer by using either tools provided within MetaboAnalysts suite (“Enrichment analysis” and “Pathway analysis tool”) or LION/Web, however we did not obtain any meaningful results. The results obtained from the MetaboAnalyst Enrichment analysis and Pathway analysis tools provided only limited information, which is

also redundant and/or less informative when compared to the other visualization tools already used in the manuscript. Two example images below represent the outcomes of the enrichment analysis using MetaboAnalyst, illustrating significant enrichment of sphingolipids in fibrotic female TAV vs male TAV. This information is already provided both by the volcano plots (Figure 3f-h), previously used PCA, as well as newly provided heatmaps (Figure 3e, Supplementary Figure 5) but with a higher level of molecular species resolution.

Metabolite Set	Total	Hits	Expect	P value	Holm P	FDR	Details
Sphingolipid metabolism	32	2	0.0832	0.00245	0.196	0.196	View
Glycerophospholipid metabolism	36	1	0.0936	0.0904	1.0	1.0	View
Steroid biosynthesis	41	1	0.107	0.102	1.0	1.0	View

Similar results were obtained using LION/Web tool enrichment (see exemplary image below, generated for fibrotic female TAV vs male TAV data). Additionally we fail to explain any functional meaning for the enrichment of endosomal/lysosomal pathway in our biological context beyond simple enrichment of SP in our datasets.

Based on the fact that enrichment analysis did not provide any additional insights or information relative to the already presented data, we opted not to include it in the revised version of the manuscript.

a) Filling missing values with "1/5 of the minimum value of the corresponding lipid across all samples ..." is in my assessment not the right way and it has a bias. Please state from which occupation threshold (% samples with experimental value) you have filled in missing values? Please use a random forest model for comparison.

We would like to point out that that quantification was performed only for the lipids showing linear MS response. Thus, out of 1073 identified lipids, only 480 were quantified. For the lipids used for quantification, the missing value count is 39 out of 24225 values (0.16%) and the missing values distributed in 25 lipid species from 6 samples. We use 1/5 of the minimum as a zero filling algorithm. We understand that it would be more accurate to perform the check for Missing Completely at Random (MCAR), Missing at Random (MAR), and Missing Not at Random (MNAR) and apply different zero filling methods in a complex study. However, in this study we have very low missing value ratio and application of more complex algorithms might be not necessary. Due to the number of available replicates, the sample size is not sufficient for a random forest model training and fitting analysis. With this sample size, it is likely to obtain a biased overfitting

model, thus we would like to apply random forest / support vector machines (SVM) and other machine learning methods when a larger dataset becomes available.

3. I find many of the figures lack annotations that help the reader recognizing certain claims in the text. Specifically, I would appreciate usage of fatty acyl/alkyl level annotation where possible. Figure 3g-i / Figure 4 a-c,e. For lipids with significant changes some representative should be chosen and shown.

Thank you for this comment. Indeed, the previous way of the presentation was not fully informative. In the revised version of the manuscript, we provide updated images of volcano plots (now Figure 3f, g, h and Figure 4a, b, c; see above) showing annotations for significantly regulated representative lipids. All lipids, except TG, are reported at the level of molecular species (listing corresponding acyl/alkyl chains). TG lipids were quantified at the level of lipid species due to the fact that isomeric species are not always chromatographically resolved sufficiently well to allow for an accurate peak integration.

To further improve the presentation of the results, in the revised version of the manuscript we included Supplementary Tables 4 and 5, reporting significantly regulated lipids used for the visualization via heatmaps and volcano plots represented on Figures 3 and 4. Previously, these data were reported in the Source Data files. We decided to provide them now as separate Supplementary Tables so readers can easily find, access and explore lists of regulated lipids.

Supplementary Table 4 | One-way ANOVA & post-hoc results comparing TAV and BAV lipidomes along different stages of FCAVD development. **a**, statistical comparison of mildly diseased, fibrotic and calcified lipidomes of TAV; **b**, statistical comparison of male and female mildly diseased, fibrotic and calcified lipidomes of TAV; **c**, statistical comparison of male and female mildly diseased, fibrotic and calcified lipidomes of TAV and BAV.

Supplementary Table 5 | Pairwise comparison of significantly regulated lipids (determined by Fold Change (FC) Analysis > 1.5 and T-test $P < 0.05$) between (a) female versus male sex in TAV tissue at mildly diseased, fibrotic stage and calcific stage as well as between (b) TAV versus BAV morphology at mildly diseased, fibrotic and calcific stage.

Reviewer: 2

The authors performed quantitative lipidomics to profile the metabolic trajectories in human tricuspid and bicuspid aortic valves, and found stage dependent extrinsic and intrinsic lipid trends. Major extrinsic lipids derived from infiltrating lipoproteins and were further metabolized within the AV. Intrinsic lipid remodeling suggested tissue degeneration with a loss of phosphatidylserines. Male and female patients showed markedly different lipid signatures of FCAVD progression. The high extent of sexual dimorphism in the valve lipidome strongly suggests that tailored approaches for pharmacological intervention. This is an interesting study with a considerable amount of work on an important and timely topic. There are however several issues to consider:

1- The authors report substantial differences between BAV vs. TAV and also between men vs. women. However, as shown in Supplementary Table 1, these results are derived from small subgroups ($n < 15$ per group and < 10) and with important differences in age and most likely in comorbidities and cardiovascular risk factors (not shown) between groups. Age is a major confounding factor in these analyses. Age likely has a major effect on lipidomics and valve lipidome in particular. There is about 10 years difference between BAV and TAV. So it is impossible to determine to which extent the differences in valve lipidome between BAV vs. TAV is related or not to age.

We sincerely thank the Reviewer for thoughtful comments and for highlighting the critical aspects related to subgroup size, age differences, and potential confounding factors such as comorbidities and cardiovascular risk. We agree that these are essential considerations when interpreting our results. Regarding the subgroup sizes, we acknowledge that the numbers are indeed not large, however, we provide a sample size of 27 aortic valves from females and 28 from males as well as 30 tricuspid and 25 bicuspid aortic valves, which we feel provides sufficient statistical strength and power to draw conclusions. The obvious age difference between bicuspid and tricuspid aortic valves at the time of replacement aligns with previous studies and also clinical experience in that a bicuspid valve becomes stenotic and symptomatic approximately one decade before a stenotic tricuspid valve. This age difference is inherent to the disease and matching for age would select for a subpopulation of patients with bicuspid valves with late onset of disease that may not represent the overall cohort of patients with bicuspid aortic valves.

While we recognize and appreciate the validity of the Reviewer concerns, we believe our results remain robust and provide valuable insights into the lipidomic differences between bicuspid and tricuspid aortic valves, as well as between men

and women. We have endeavored to transparently discuss these limitations in the manuscript to provide a balanced interpretation of our findings.

In Results section: "It is important to note that individuals with BAVs typically experience an earlier onset of FCAVD, with clinically manifested symptomatic AS occurring on average 10 years earlier than in patients with the physiological TAV. Subsequently, the observed variability in lipidomes between TAV and BAV may be attributed to differences in the age of patients with bicuspid versus tricuspid aortic valves."

In the Discussion section: „Interestingly, metabolic trajectories of BAV along FCAVD development were less sex-specific and generally more similar to the female TAV, thus associated with higher SP accumulation, possibly reflecting a prevalence of the fibrotic stage. It can also be interpreted that the hemodynamic effects are more detrimental for the pathological process in BAV, rather than the sex-specific mechanisms that may be more relevant in TAV. Further, age differences between TAV and BAV FCAVD patients might be responsible for some of the observed differences in AV lipidomes."

In addition, in the revised manuscript, we provide a more detailed Supplementary Table 1 with the relevant baseline clinical parameters of the tissue donors.

Supplementary Table 1 | Clinical metadata. a, Number of samples per group. Median [IQR] or total count per group (%). **b,** Baseline parameters per sex in tricuspid aortic valves. Median [IQR] or total count per group (%). P values were calculated by T-test. **c,** Baseline parameters per sex in all patients. Median [IQR] or total count per group (%). P values were calculated by T-test. **d,** Baseline parameters per valve morphology. Median [IQR] or total count per group (%). P values were calculated by T-test.

Supplementary Data Table 1a | Number of samples per group
Median [IQR] or total count per group (%).

Pathology	Morphology	Sex	N
Mildly diseased	TAV	male	13
		female	10
	BAV	male	5
		female	6
Fibrotic	TAV	male	15
		female	13
	BAV	male	11
		female	12
Calcific	TAV	male	14
		female	13
	BAV	male	11
		female	10

Supplementary Data Table 1b | Baseline parameters per sex in tricuspid aortic valves
Median [IQR] or total count per group (%). P values were calculated by T-test.

	Female (14)	Male (16)	P-value
Age	71 [69; 73]	72 [67; 76]	0.57
BMI	30 [27; 33]	27 [25; 33]	0.30
Indication - AS III*	14 (100)	16 (100)	1.00
Diabetes mellitus	3/14 (21)	7/16 (44)	0.26
Dialysis	1/14 (7)	0/16 (0)	0.99
Total cholesterol	5.3 [4.8; 5.7]	5.2 [4.6; 5.6]	0.77
LDL-cholesterol mmol/L	3.6 [2.8; 3.9]	4.2 [3.3; 4.4]	0.33
HDL-cholesterol mmol/L	1.5 [1.2; 1.7]	1.3 [1.2; 1.6]	0.85
TG mmol/L	0.9 [0.8; 1.6]	1.5 [1.3; 1.6]	0.62
Hyperlipidemia	9/14 (64)	14/16 (88)	0.20
Statin therapy	9/14 (64)	13/16 (81)	0.41
Coronary artery disease	7/12 (58)	7/16 (44)	0.70
Arterial hypertension	11/14 (79)	14/16 (88)	0.64
ACE-inhibitor therapy	09/14 (64)	12/16 (75)	0.60
Estimated glomerular filtration rate (GFR/mL/min/1.73m ²)	72 [51; 82]	75 [67; 89]	0.16
Transvalvular gradients (Pmax/mmHg)	62 [58; 73]	64 [54; 78]	0.37
Transvalvular gradients (mean/mmHg)	37 [32; 50]	40 [37; 44]	0.60
Stroke Volume Index (SVI)	44 [42; 45]	39 [32; 43]	0.40
Left ventricular ejection fraction (LVEF)	61 [56; 69]	57 [50; 63]	0.12

Supplementary Data Table 1c | Baseline parameters per sex in all patients
Median [IQR] or total count per group (%). P values were calculated by T-test.

	Female (27)	Male (28)	P-value
Age	70 [65; 73]	68 [58; 73]	0.39
BMI	29 [25; 31]	28 [25; 32]	0.48
Indication - AS III*	27 (100)	28 (100)	1.00
Diabetes mellitus	3/27 (10)	8/27 (30)	0.09
Dialysis	1/27 (4)	0/27 (0)	0.98
Total cholesterol	5.5 [4.8; 5.9]	5.9 [5.3; 6.1]	0.52
LDL-cholesterol mmol/L	3.4 [2.3; 4.3]	3.7 [3.2; 4.2]	0.92
HDL-cholesterol mmol/L	1.7 [1.3; 2.7]	1.3 [1.0; 1.6]	0.03
TG mmol/L	1.6 [1.3; 2.0]	2.2 [0.9; 2.3]	0.36
Hyperlipidemia	16/26 (62)	22/28 (79)	0.24
Statin therapy	17/27 (63)	19/28 (68)	0.78
Coronary artery disease	12/25 (48)	12/28 (43)	0.79
Arterial hypertension	18/27 (67)	24/28 (86)	0.12
ACE-inhibitor therapy	17/27 (63)	20/28 (71)	0.20
Estimated glomerular filtration rate (GFR/mL/min/1.73m ²)	75 [60; 85]	84 [73; 88]	0.11
Transvalvular gradients (Pmax/mmHg)	66 [55; 83]	69 [59; 87]	0.29
Transvalvular gradients (mean/mmHg)	46 [35; 53]	41 [37; 52]	0.63
Stroke Volume Index (SVI)	41 [38; 43]	41 [30; 43]	0.63
Left ventricular ejection fraction (LVEF)	61 [55; 68]	56 [49; 63]	0.06

Supplementary Data Table 1d | Baseline parameters per valve morphology
Median [IQR] or total count per group (%). P values were calculated by T-test.

	TAV (30)	BAV (25)	P-value
Age	71 [67; 76]	65 [58; 70]	<0.01
BMI	28 [25; 30]	29 [26; 33]	0.13
Indication - AS III*	30 (100)	25 (100)	1.00
Diabetes mellitus	10/30 (33)	1/25 (4)	0.01
Dialysis	1/30 (3)	0/25 (0)	0.99
Total cholesterol	5.1 [4.6; 5.5]	5.9 [5.8; 6.1]	0.10
LDL-cholesterol mmol/L	3.6 [2.9; 4.4]	3.5 [2.9; 4.2]	0.50
HDL-cholesterol mmol/L	1.4 [1.2; 1.6]	1.4 [1.0; 2.1]	0.12
TG mmol/L	1.3 [1.2; 1.6]	2.1 [2.0; 2.7]	0.07
Hyperlipidemia	23/30 (77)	15/24 (63)	0.37
Statin therapy	22/30 (73)	14/25 (56)	0.26
Coronary artery disease	14/28 (50)	12/25 (48)	1.00
Arterial hypertension	25/30 (83)	17/25 (68)	0.21
ACE-inhibitor therapy	22/30 (73)	15/25 (60)	0.45
Estimated glomerular filtration rate (GFR/mL/min/1.73m ²)	74 [58; 85]	84 [74; 89]	0.09
Transvalvular gradients (Pmax/mmHg)	64 [56; 77]	79 [56; 89]	0.37
Transvalvular gradients (mean/mmHg)	39; 34; 49	46 [43; 60]	0.01
Stroke Volume Index (SVI)	41 [35; 44]	41 [31; 43]	0.71
Left ventricular ejection fraction (LVEF)	60 [53; 64]	60 [53; 65]	0.74

2- It is likely that men and women subgroups were not well balanced in terms of age and other cardiometabolic risk factors, which may have also affected the results on sex related differences. The authors would need to add a comprehensive table showing and comparing the age and other baselines characteristics of the patients in the different subgroups. This table should include, age, sex, BMI, dyslipidemia, hypertension, diabetes, coronary artery disease, pharmacotherapy at the time of valve explant (statins, anti-hypertensive therapy etc.).

Thank you for your insightful comment regarding the potential influence of age and cardiometabolic risk factors on the observed sex-related differences. We appreciate your suggestion to provide a comprehensive summary of baseline characteristics to enhance the clarity and context of our findings. In response to your suggestion, we have now included a detailed table summarizing the clinical parameters for each subgroup – see above and also as updated **Supplementary Table 1**. This table provides information on age, sex, BMI, dyslipidemia,

hypertension, diabetes, coronary artery disease, and pharmacotherapy at the time of valve explant (e.g., use of lipid-lowering medication and antihypertensive medications). Additionally, we have incorporated other relevant clinical parameters.

We believe that these additions will help contextualize our results and allow readers to better assess the potential impact of these factors on the observed differences. Of note, the only statistically significant difference between females and males was in high density lipoprotein cholesterol levels, which is consistent with the literature, that females tend to have higher HDL serum levels.

3- Pharmacotherapy that was present at the time of valve explant may have affected the results of valve lipidome, and in particular therapies targeting lipids and diabetes. This information was not provided in the paper and it is likely that the different BAV vs. TAV, mean vs. women and stage of disease were not well matched for age, cardiometabolic risk factors, and pharmacotherapies.

We would like to emphasize that we paid special attention to ensuring even distribution of clinical parameters when designing the study protocol. As a result, most clinical parameters were well balanced across the groups, minimizing potential confounding effects. However, we do acknowledge that patients with tricuspid aortic valves had a higher prevalence of diabetes mellitus as compared to bicuspid aortic valves. In addition, there was a clinically expected age difference between bicuspid and tricuspid aortic valve patients for the points mentioned in the first response to this Reviewer's concerns. Our clinical parameter inventory was limited to lipid lowering medication and antihypertensive medications. Anti-diabetic therapy was not specifically recorded.

4- The methods on the lipidomic analyses is very well described and detailed. On the other hand, the section on Human aortic valve tissue samples is described briefly in a short paragraph. Still this sections is absolutely essential. The matching of the comparison groups for age, cardiometabolic risk factors, and pharmacotherapies is key for the validity and the interpretability of the inter-group comparisons.

We appreciate this Reviewer's emphasis on the importance of the section on human aortic valve tissue samples, as well as the critical role of matching the comparison groups for baseline parameters to ensure the validity and interpretability of inter-group comparisons. Following the Reviewer suggestion, we have expanded the section on human aortic valve tissue samples in the revised manuscript. The updated section now includes detailed descriptions of how the samples were obtained. No specific inclusion and exclusion criteria were applied,

and no additional steps had to be taken to match groups for key variables such as age, cardiometabolic risk factors, and pharmacotherapies, as except for the mentioned differences in the prevalence of diabetes, baseline clinical parameters were evenly distributed among the study groups and we did not intend to correct for the age differences between tricuspid and bicuspid valve donors as this difference is inherent to the disease itself. We ensured careful consideration of these factors during study design to minimize confounding and improve the robustness of our findings.

„Human aortic valve tissue samples

FCAVD tissues were obtained from 55 patients undergoing aortic valve replacement for severe AS (Supplementary Table 1). Tissue collection was approved by the local Ethical Committee (Medical Faculty, University Leipzig, registration number 128/19-ek) and all patients gave written informed consent in accordance with the Declaration of Helsinki. Using a stereoscopic microscope, tissue samples were macroscopically inspected and dissected into mildly diseased, fibrotic and calcific sections on-site and flash frozen in liquid nitrogen (Fig. 2a, Supplemental Fig. 7). The tissue allocation was confirmed histologically for an initial protocol establishment phase using Movat pentachrome staining as well as molecular imaging with a bisphosphonate based calcium tracer (OsteoSense) to visualize and confirm fibrosis and calcification. Deep-frozen tissue sections were grinded in a percussion mortar in liquid nitrogen and stored at -80 °C until further analysis.“

5- The methods used to classify determine the status of the lesion: mildly diseased, fibrotic and calcific is not described.

We thank this Reviewer for providing us with the opportunity to further explain our methodology. Specific additions were made to the methods section as highlighted above. Additionally, we introduced a new Supplementary Figure 7 to illustrate visually the selection of the corresponding AV sections.

Supplementary Fig. 7 | Visual representation of aortic valves sectioning. AV were macroscopically inspected and dissected into mildly diseased (blue area), fibrotic (yellow area) and calcific (violet area) sections.

Sectioned by disease stage

Mildly diseased

Fibrotic

Calcific

6- The clinical implications of the results of this study are unclear and would merit to be better described.

We thank this Reviewer for encouraging us to elaborate more on the clinical implications of our study. Our study uncovers distinct lipidomic signatures in FCAVD, identifying both lipoprotein-driven and valve-intrinsic lipid alterations. In the fibrotic stage, major membrane-building lipids accumulate, reflecting increased cell populations, while advanced calcification is marked by oxidative degradation of PUFA etherPE and depletion of PS lipids, potentially contributing to calcification through hydroxyapatite nucleation. We observed sex-specific lipidomic trajectories, with male valves showing higher PUFA TG and female valves exhibiting elevated SM and Cer lipids, linked to fibrosis severity. These differences align with TGF- β signaling, which is upregulated in women, suggesting a mechanistic link between Cer metabolism and fibrosis. Additionally, bicuspid aortic valve lipid trajectories were less sex-specific and more aligned with female tricuspid aortic valves, highlighting the dominant role of hemodynamic stress in BAV pathology. The identification of ceramide accumulation as a key factor in FCAVD progression offers opportunities for pharmacological interventions targeting Cer biosynthesis or degradation. The study also reveals the interplay between systemic lipid metabolism and valvular lipid homeostasis, suggesting that sphingolipids may play a role in FCAVD pathogenesis. Importantly, these sex-specific differences in lipidomic trajectories underscore the need for greater inclusion of biological sex in FCAVD research and clinical trials, which is crucial for improving diagnostic accuracy and developing tailored therapeutic strategies. Overall, our study provides insights into the sexually dimorphic lipidome of FCAVD, particularly in female pathogenesis, and highlights potential therapeutic targets to

mitigate fibrosis and calcification. We rewrote the concluding part of the discussion to specifically highlight the major novel findings of our analysis.

"In conclusion, this study identified distinct lipidomic signatures in FCAVD and provides a molecular lipid map of the AV, highlighting both lipoprotein-derived and valve-intrinsic lipid alterations. The identification of Cer accumulation as a key factor in fibrotic FCAVD progression, which represents the dominant phenotype in female FCAVD, opens avenues for the development of pharmacological interventions aimed at reducing Cer levels, either through inhibition of their biosynthesis or enhancement of their degradation.^{34,70,71} Thus, targeting SP rather than cholesterol metabolism might be a promising option for FCAVD treatment. Importantly, we identified significant sex-specific responses in FCAVD progression. With women presenting significantly higher levels of AV Cer, future development in SP-based diagnostics and therapeutic interventions should be performed in a sex-specific manner. Indeed, the general misperception that women are less susceptible to CVD stems from an under-representation of women in clinical trials. Even in cohorts used for the validation of CERT or sphingolipid-inclusive score (SIC), as strong CVD event predictors, 72% and 77% of the cohorts, respectively, were represented by male individuals^{72,73}. The sex-specific differences in FCAVD identified in this study call for greater inclusion of women in research and clinical trials to improve diagnostic accuracy and develop tailored therapeutic strategies that address these disparities. Furthermore, the study reveals the interplay between systemic lipid metabolism and valvular lipid homeostasis, suggesting that circulating sphingolipids may play a role in FCAVD pathogenesis. These insights strengthen the rationale for investigating organ-to-organ metabolic cross-talk in FCAVD."